# Near-Optimal No-Regret Learning Dynamics for General Convex Games

**Gabriele Farina**[*]
Carnegie Mellon University
Pittsburgh, PA 15213
gfarina@cs.cmu.edu

**Ioannis Anagnostides**[*]
Carnegie Mellon University
Pittsburgh, PA 15213
ianagnos@cs.cmu.edu

**Haipeng Luo**
University of Southern California
Los Angeles, CA 90007
haipengl@usc.edu

**Chung-Wei Lee**
University of Southern California
Los Angeles, CA 90007
leechung@usc.edu

**Christian Kroer**
Columbia University
New York, NY 10027
christian.kroer@columbia.edu

**Tuomas Sandholm**
Carnegie Mellon University
Strategy Robot, Inc.
Optimized Markets, Inc.
Strategic Machine, Inc.
Pittsburgh, PA 15213
sandholm@cs.cmu.edu

## Abstract

A recent line of work has established uncoupled learning dynamics such that, when employed by all players in a game, each player's *regret* after $T$ repetitions grows polylogarithmically in $T$, an exponential improvement over the traditional guarantees within the no-regret framework. However, so far these results have only been limited to certain classes of games with structured strategy spaces—such as normal-form and extensive-form games. The question as to whether $O(\mathrm{polylog}\, T)$ regret bounds can be obtained for general convex and compact strategy sets—which occur in many fundamental models in economics and multiagent systems—while retaining efficient strategy updates is an important question. In this paper, we answer this in the positive by establishing the first uncoupled learning algorithm with $O(\log T)$ per-player regret in general *convex games*, that is, games with concave utility functions supported on arbitrary convex and compact strategy sets. Our learning dynamics are based on an instantiation of optimistic follow-the-regularized-leader over an appropriately *lifted* space using a *self-concordant regularizer* that is peculiarly not a barrier for the feasible region. Our learning dynamics are efficiently implementable given access to a proximal oracle for the convex strategy set, leading to $O(\log \log T)$ per-iteration complexity; we also give extensions when access to only a *linear* optimization oracle is assumed. Finally, we adapt our dynamics to guarantee $O(\sqrt{T})$ regret in the adversarial regime. Even in those special cases where prior results apply, our algorithm improves over the state-of-the-art regret bounds either in terms of the dependence on the number of iterations or on the dimension of the strategy sets.

---

[*]Equal contribution.

36th Conference on Neural Information Processing Systems (NeurIPS 2022).

# 1 Introduction

*Regret minimization* is a celebrated framework that has been central in the development of online learning and the theory of multiagent systems. Indeed, fundamental connections have been forged between no-regret learning and game-theoretic solution concepts [Freund and Schapire, 1999, Hart and Mas-Colell, 2000, Foster and Vohra, 1997, Roughgarden, 2015]. More broadly, regret is an intrinsic measure of performance in online learning and games. Furthermore, regret minimization algorithms have enjoyed a remarkable practical success, being a primary component in recent landmark results in AI [Bowling et al., 2015, Moravčík et al., 2017, Brown and Sandholm, 2017, 2019]. These advances were guided by game-theoretic principles, made possible by training the AI agents using *self-play* under regret-minimizing algorithms, an approach that has proven to be more scalable compared to linear programming techniques. Nevertheless, the traditional no-regret framework is overly pessimistic, insisting on modeling the environment in a fully adversarial way. While this well-understood worst-case view might be justifiable for applications such a security games, it could be far from optimal in more benign and *predictable* environments, including the setting of training agents using self-play. This begs the question: *What are the optimal performance guarantees we can obtain when learning agents are competing against each other in general games?*

This fundamental question was first formulated and addressed by Daskalakis et al. [2011] within the context of *zero-sum games*. Since then, there has been a considerable interest in extending their guarantee to more general settings [Rakhlin and Sridharan, 2013, Syrgkanis et al., 2015, Foster et al., 2016, Chen and Peng, 2020, Daskalakis and Golowich, 2022, Piliouras et al., 2022]. In particular, Daskalakis et al. [2021] recently established that when all players in a general *normal-form game* employ an *optimistic* variant of *multiplicative weights update (MWU)*, the regret of each player grows *nearly-optimally* as $O(\log^4 T)$ after $T$ repetitions of the game, leading to an *exponential improvement* over the guarantees obtained using traditional techniques within the no-regret framework. However, while normal-form games are a common way to represent strategic interactions in theory, most settings of practical significance inevitably involve more complex strategy spaces. For those settings, any faithful approximation of the game using the normal form is typically inefficient, requiring an action space that is exponential in the natural parameters of the problem, thereby limiting the practical implications of those prior results. This motivates our central question:

> *Can we establish near-optimal, efficiently implementable, and strongly uncoupled no-regret learning dynamics in general convex games?* (♣)

*Convex games* are a rich class of games wherein the strategy space of each player is an arbitrary convex and compact set, while the utility of each player is an arbitrary concave function (see Section 2 for a formal description). As such, convex games encompass normal-form and *extensive-form games*, but go well-beyond to many other fundamental settings in economic theory including routing games, resource allocation problems, and competition between firms. Our primary contribution in this paper is to substantially extend prior results to all such games, addressing Question (♣).

## 1.1 Our Contributions

In this paper we introduce a novel no-regret learning algorithm, which we coin *lifted log-regularized optimistic follow the regularized leader (LRL-OFTRL)*. LRL-OFTRL settles Question (♣) in the positive, as summarized in the following theorem.[2]

**Theorem 1** (Detailed version in Theorem 4)**.** *Consider any general convex game. When all players employ our strongly uncoupled learning dynamics (LRL-OFTRL), the regret of each player grows as $O(\log T)$. At the same time, if the player is facing adversarial utilities we guarantee $O(\sqrt{T})$ regret.*

Importantly, our learning dynamics are efficiently implementable given access to a *proximal oracle* for the set (Equation (7)), requiring only $O(\log \log T)$ operations per iteration (Theorem 5); such an oracle is weaker than the—relatively standard in convex optimization—*quadratic optimization oracle*. We also point out extensions under a weaker *linear optimization oracle*, albeit with a worse per-iteration complexity (Theorem 6). Our no-regret learning dynamics imply the first efficiently implementable and near-optimal regret guarantees in general convex games, significantly extending

---

[2]For simplicity in the exposition we use the $O(\cdot)$ notation in our introduction to suppress time-independent parameters that depend (polynomially) on the game; precise statements are deferred to Section 3.

| Method | Applies to | Regret bound | Cost per iteration |
|---|---|---|---|
| OFTRL / OMD [Syrgkanis et al., 2015] | General convex set | $O(\sqrt{n}\,\mathfrak{R}T^{1/4})$ | Regularizer- & oracle- dependent |
| OMWU [Daskalakis et al., 2021] | Simplex $\Delta^d$ | $O(n\log d\log^4 T)$ | $O(d)$ |
| Clairvoyant MWU [Piliouras et al., 2022] | Simplex $\Delta^d$ | $O(n\log d)$ Subsequence only ‡ | $O(d)$ |
| Kernelized OMWU [Farina et al., 2022] | Polytope $\Omega = \mathrm{co}\mathcal{V}$ with $\mathcal{V} \subseteq \{0,1\}^d$ | $O(n\log|\mathcal{V}|\log^4 T)$ | $d \times$ cost of kernel |
| LRL-OFTRL [This paper] | General convex set $\mathcal{X} \subseteq \mathbb{R}^d$ | $O(nd\|\mathcal{X}\|_1^2\log T)$ | Oracle-dependent: 
 • $O(\log\log T)$ proximal oracle calls 
 • $O(\mathrm{poly}\,T)$ linear opt. oracle calls |

Table 1: Comparison of prior results on minimizing external regret in games. For simplicity, we have suppressed dependencies on the smoothness and the range of the utilities. We use $n$ to denote the number of players; $T$ to denote the number of repetitions; $\mathfrak{R}$ to indicate a parameter that depends on the regularizer; $\mathrm{co}\mathcal{V}$ to denote the convex hull of $\mathcal{V}$; and $\|\mathcal{X}\|_1$ to denote a bound on the maximum $\ell_1$ norm of any strategy. ‡ Unlike all other algorithms, the full sequence of iterates produced by Clairvoyant MWU (CMWU) is not known to achieve sublinear regret. Rather, after running CMWU for $T$ iterations, only a smaller subsequence of length $\Theta(T/\log T)$ iterates is known to attain the regret stated in the table. So, we remark that in order to achieve a comparable approximation of a coarse correlated equilibrium, CMWU needs to be run for $\Theta(T\log T)$ iterations.

the scope of prior $O(\mathrm{polylog}\,T)$-regret guarantees [Daskalakis et al., 2021, Farina et al., 2022]; a comparison with prior approaches is included in Table 1. We remark that Theorem 1 establishes near-optimal regret both under *self-play*, and in the adversarial regime—meaning that the other players act so as to minimize the player's utility; the latter feature of adversarial robustness has been a central desideratum in this line of work (*e.g.*, see the discussion in [Kangarshahi et al., 2018, Daskalakis et al., 2011]).

Our proposed learning dynamics lie within the general framework of *optimistic* no-regret learning, pioneered by Chiang et al. [2012] and Rakhlin and Sridharan [2013]. We leverage the OFTRL algorithm of Syrgkanis et al. [2015], but with some important twists. First, as detailed in Algorithm 1, the OFTRL optimization step is performed over a "lifted" space. While prior work in online learning has employed similar in spirit approaches [Lee et al., 2020, Luo et al., 2022], our lifting is quite different, ensuring that the regret incurred by OFTRL is *nonnegative* (Theorem 2). Further, we employ a *logarithmic self-concordant regularizer*; interestingly, and perhaps surprisingly, this is not a *barrier* for the underlying feasible set. This deviates substantially from the typical use of self-concordant regularization (especially within the bandit setting [Abernethy et al., 2008, Wei and Luo, 2018, Bubeck et al., 2019]). A pictorial overview of our construction is given in the caption of Algorithm 1.

The use of the logarithmic regularizer serves two main purposes. First, we show that it guarantees *multiplicative stability* of the strategies, a refined notion of stability that is also leveraged in the work of Daskalakis et al. [2021]. Nonetheless, we are the first to leverage such properties in general domains, going well beyond the guarantees of (Optimistic) MWU on the simplex [Daskalakis et al., 2021]. Further, the *local norm* induced by the logarithmic regularizer enables us to cast regret bounds from the lifted space to the original space, while preserving the *RVU property* [Syrgkanis et al., 2015, Definition 3]. In turn, this implies near-optimal regret by establishing that the *second-order path lengths* up to time $T$ are bounded by $O(\log T)$ (Theorem 3), building on a recent technique of Anagnostides et al. [2022a] which crucially leverages the nonnegativity of swap regret.[3]

---

[3]To see why nonnegativity is crucial, note that the RVU bound implies optimal *sum* of players' regrets [Syrgkanis et al., 2015]. Thus, nonnegativity would imply the same bound for each player's regret.

## 1.2 Further Related Work

The rich line of work pursuing improved regret guarantees in games was pioneered by Daskalakis et al. [2011]. Specifically, they developed *strongly uncoupled* learning dynamics so that the players' regrets grow as $O(\log T)$, an exponential improvement over the guarantee one could hope for in adversarial environments [Shalev-Shwartz, 2012, Cesa-Bianchi and Lugosi, 2006]. Their result was significantly simplified by Rakhlin and Sridharan [2013]—again in zero-sum games—who introduced a simple variant of *mirror descent* with a *recency bias*—a.k.a. *optimistic* mirror descent (OMD). It is worth noting that, beyond the benefits of optimism from an optimization standpoint [Polyak, 1987], recency bias has been experimentally documented in natural learning environments in economics [Fudenberg and Peysakhovich, 2014].

Subsequently, Syrgkanis et al. [2015] crystallized the *RVU property*, an adversarial regret bound applicable for a broad class of optimistic no-regret learning algorithms. Using that property, they showed that the individual regret of each player grows as $O(T^{1/4})$ in general games, thereby converging to the set of *coarse correlated equilibria* with a rate of $O(T^{-3/4})$. A near-optimal bound of $O(\mathrm{polylog}(T))$ in normal-form games was finally established by Daskalakis et al. [2021], while Farina et al. [2022] generalized that result in a class of polyhedral games that includes extensive-form games. Some extensions of the previous results have also been established for the stronger notion of *no-swap-regret* learning dynamics in normal-form games [Chen and Peng, 2020, Anagnostides et al., 2022b,a]. In particular, our work builds on a very recent technique of Anagnostides et al. [2022a], which established $O(\log T)$ swap regret in normal-form games using as a regularizer a self-concordant *barrier* function. On the other hand, establishing even sublinear $o(T)$ swap regret in extensive-form games is a notorious open question. Finally, an interesting new approach for obtaining near-optimal external regret in normal-form games was recently proposed in concurrent work by Piliouras et al. [2022].[4]

Games with continuous strategy spaces have received a lot of attention in the literature; *e.g.*, see [Roughgarden and Schoppmann, 2015, Even-Dar et al., 2009, Harks and Klimm, 2011, Hsieh et al., 2021, Mertikopoulos and Zhou, 2019, Stein et al., 2011, Stoltz and Lugosi, 2007], and references therein. Such games encompass a wide variety of applications in economics and multiagent systems; we give several examples in Section 2. Indeed, in many applications of interest a faithful approximation of the game requires an extremely large or even infinite action space; such settings could be abstracted as *Littlestone games* in the sense of the recent work of Daskalakis and Golowich [2022].

## 2 No-Regret Learning and Convex Games

In this section we review the general setting of *convex games*[5] which encompasses a number of important applications, as explained in Section 2.2. We then formally define the framework of uncoupled and online no-regret learning in games in Section 2.3.

**Notation**    We let $\mathbb{N} = \{1, 2, \ldots, \}$ be the set of natural numbers. For a vector $\boldsymbol{x} \in \mathbb{R}^d$ we denote by $\boldsymbol{x}[r]$ its $r$-th coordinate, for some index $r \in [\![d]\!] := \{1, 2, \ldots, d\}$. We will typically represent the players using subscripts; superscripts are reserved for the time index, denoted by the variable $t$.

### 2.1 Convex Games

Let $[\![n]\!] := \{1, 2, \ldots, n\}$ be a set of players, with $n \in \mathbb{N}$. In a *convex game*, every player $i \in [\![n]\!]$ has a nonempty convex and compact set of strategies $\mathcal{X}_i \subseteq \mathbb{R}^{d_i}$. For a *joint strategy profile* $\boldsymbol{x} = (\boldsymbol{x}_1, \ldots, \boldsymbol{x}_n) \in \bigtimes_{j=1}^{n} \mathcal{X}_j$, the reward of player $i$ is given by a continuously differentiable utility function $u_i : \bigtimes_{j=1}^{n} \mathcal{X}_j \to \mathbb{R}$ subject to the following standard assumption.

---

[4]An earlier version of the paper [Piliouras et al., 2021] proposed a preliminary and *not uncoupled* version of the Clairvoyant MWU algorithm whose iterates were guaranteed to be no-regret and require $O(d \log T)$ per-iteration complexity. The 2022 revision of that paper provides an *uncoupled* version with time-independent $O(d)$ per-iteration complexity, albeit at the cost of losing the no-regret guarantee on the entire sequence of iterates. See also footnote ‡ in Table 1.

[5]Sometimes these are referred to as *concave games* [Rosen, 1965] or *continuous games* [Hsieh et al., 2021].

**Assumption 1** (Convex games). *The utility function $u_i(\boldsymbol{x}_1, \ldots, \boldsymbol{x}_n)$ of any player $i \in [\![n]\!]$ satisfies the following properties:*

1. *(Concavity) $u_i(\boldsymbol{x}_i, \boldsymbol{x}_{-i})$ is* concave *in $\boldsymbol{x}_i$ for $\boldsymbol{x}_{-i} = (\boldsymbol{x}_1, \ldots, \boldsymbol{x}_{i-1}, \boldsymbol{x}_{i+1}, \ldots, \boldsymbol{x}_n) \in \bigtimes_{j \neq i} \mathcal{X}_j$;*
2. *(Bounded gradients) for any $(\boldsymbol{x}_1, \ldots, \boldsymbol{x}_n) \in \bigtimes_{j=1}^{n} \mathcal{X}_j$, $\|\nabla_{\boldsymbol{x}_i} u_i(\boldsymbol{x}_1, \ldots, \boldsymbol{x}_n)\|_\infty \leq B$, for some parameter $B > 0$; and*
3. *(L-smoothness) there exists $L > 0$ so that for any two joint strategy profiles $\boldsymbol{x}, \boldsymbol{x}' \in \bigtimes_{j=1}^{n} \mathcal{X}_j$,*

$$\|\nabla_{\boldsymbol{x}_i} u_i(\boldsymbol{x}) - \nabla_{\boldsymbol{x}_i} u_i(\boldsymbol{x}')\|_\infty \leq L \sum_{j \in [\![n]\!]} \|\boldsymbol{x}_j - \boldsymbol{x}'_j\|_1.$$

## 2.2 Applications and Examples of Convex Games

Here we discuss several different classes of games which can all be analyzed under the common framework of convex games. For simplicity, we describe Cournot competion in the one-dimensional setting, but it can be readily generalized in more general domains. For more examples, we refer to [Even-Dar et al., 2009, Hsieh et al., 2021], and references therein.

**Normal-Form Games** In *normal-form games (NFGs)* every player $i \in [\![n]\!]$ has a finite and nonempty set of strategies $\mathcal{A}_i$. Player $i$'s strategy set contains all probability distributions supported on $\mathcal{A}_i$; that is, $\mathcal{X}_i = \Delta(\mathcal{A}_i)$. The utility of player $i$ can be expressed as the *multilinear* function $u_i(\boldsymbol{x}) := \mathbb{E}_{\boldsymbol{a} \sim \boldsymbol{x}}[\mathcal{U}_i(\boldsymbol{a})]$, for some arbitrary function $\mathcal{U}_i : \bigtimes_{j=1}^{n} \mathcal{A}_j \to \mathbb{R}$.

**Extensive-Form Games** *Extensive-form games (EFGs)* generalize NFGs by capturing both sequential and simultaneous moves, stochasticity from the environment, as well as *imperfect information*. EFGs are abstracted on a directed tree. Once the game reaches a terminal (or leaf) node $z \in \mathcal{Z}$, each player $i \in [\![n]\!]$ receives a utility $\mathcal{U}_i(z)$, for some $\mathcal{U}_i : \mathcal{Z} \to \mathbb{R}$. The strategy space of each player $i \in [\![n]\!]$ can be compactly represented using the *sequence-form polytope* $\mathcal{Q}_i$ [Romanovskii, 1962, Koller et al., 1996]. If $p_c(z)$ is the probability of reaching terminal node $z \in \mathcal{Z}$ over "chance moves", the utility of player $i$ can be expressed as $u_i(\boldsymbol{q}) := \sum_{z \in \mathcal{Z}} p_c(z) \mathcal{U}_i(z) \prod_{j \in [\![n]\!]} \boldsymbol{q}_j[\sigma_{j,z}]$, where $\boldsymbol{q} = (\boldsymbol{q}_1, \ldots, \boldsymbol{q}_n) \in \bigtimes_{j=1}^{n} \mathcal{Q}_j$ is the joint strategy profile, and $\boldsymbol{q}_j[\sigma_{j,z}]$ is the probability mass assigned to the last *sequence* $\sigma_{j,z}$ encountered by player $j$ before reaching $z$. The smoothness and the concavity of the utilities follow directly from multilinearity; for a more detailed account on EFGs we refer the interested reader to the excellent book of Shoham and Leyton-Brown [2008].

**Splittable Routing Games** In these games [Roughgarden and Schoppmann, 2015] every player has to route a flow $f_i$ from a source to a destination in an undirected graph $G = (V, E)$. Every edge $e \in E$ is associated with a latency function $\ell_e(f_e)$ mapping the amount of flow passing through the edge to some latency. The set of strategies of player $i$ corresponds to the possible ways of "splitting" the flow $f_i$ into paths from the source to the destination. Under suitable restrictions on the latency functions, those games satisfy Assumption 1 (see [Syrgkanis et al., 2015]).

**Cournot Competition** This game is played among $n$ firms (players). Every firm $i$ decides the *quantity* $s_i \in \mathcal{S}_i \subseteq \mathbb{R}_{\geq 0}$ of a common good to produce, where $\mathcal{S}_i$ is an interval. Further, a cost function $c_i : \mathcal{S}_i \to \mathbb{R}$ assigns a *production cost* to a given quantity, while $p : \bigtimes \mathcal{S}_i \to \mathbb{R}_{\geq 0}$ is the price of the good determined by the the joint choice of quantity $\boldsymbol{s} = (s_1, \ldots, s_n)$ across the firms. Then, the utility of firm $i$ is defined as $u_i(\boldsymbol{s}) := s_i p(\boldsymbol{s}) - c_i(s_i)$. In *linear Cournot competition*, $p(\boldsymbol{s}) := a - b \left( \sum_{i=1}^{n} s_i \right)$, for some $a, b > 0$, while the cost functions $c_i$ are assumed to be smooth and convex [Even-Dar et al., 2009].

## 2.3 Online Linear Optimization and No-Regret Learning

In the *online learning framework* a learning agent has to select a strategy $\boldsymbol{x}^{(t)} \in \mathcal{X} \subseteq \mathbb{R}^d$ at every time $t \in \mathbb{N}$. Then, in the *full information* model, the learner receives as feedback from the environment a *linear* utility function $\boldsymbol{x} \mapsto \langle \boldsymbol{x}, \boldsymbol{u}^{(t)} \rangle$, for some vector $\boldsymbol{u}^{(t)} \in \mathbb{R}^d$. The canonical measure of performance is the notion of *regret*, defined for a time horizon $T \in \mathbb{N}$ as follows.

$$\mathrm{Reg}^T := \max_{\boldsymbol{x}^* \in \mathcal{X}} \left\{ \sum_{t=1}^{T} \langle \boldsymbol{x}^*, \boldsymbol{u}^{(t)} \rangle \right\} - \sum_{t=1}^{T} \langle \boldsymbol{x}^{(t)}, \boldsymbol{u}^{(t)} \rangle. \tag{1}$$

That is, the performance of the agent is compared to the optimal *fixed* strategy in hindsight. It is important to note that *regret can be negative*. In the context of convex games, it is assumed that every player $i \in [n]$ receives at time $t$ the "linearized" utility function $\boldsymbol{x}_i \mapsto \langle \boldsymbol{x}_i, \boldsymbol{u}_i^{(t)} \rangle$, where $\boldsymbol{u}_i^{(t)} \coloneqq \nabla_{\boldsymbol{x}_i} u_i(\boldsymbol{x}^{(t)})$. By concavity (Assumption 1),

$$\max_{\boldsymbol{x}_i^* \in \mathcal{X}_i} \sum_{t=1}^{T} \left( u_i(\boldsymbol{x}_i^*, \boldsymbol{x}_{-i}^{(t)}) - u_i(\boldsymbol{x}^{(t)}) \right) \leq \max_{\boldsymbol{x}_i^* \in \mathcal{X}_i} \sum_{t=1}^{T} \langle \boldsymbol{x}_i^* - \boldsymbol{x}_i^{(t)}, \nabla_{\boldsymbol{x}_i} u_i(\boldsymbol{x}^{(t)}) \rangle.$$

As a result, a regret bound on the linearized regret—in the sense of (1)—automatically translates to a regret bound in the convex game.

**Strongly Uncoupled Learning Dynamics** In this setting, all learning dynamics are *uncoupled* in the sense of Hart and Mas-Colell [2003]: every player is oblivious to the other players' utilities. In fact, players need not have any prior knowledge about the game, even about their own utilities; this captures the condition of *strong uncoupledness* of Daskalakis et al. [2011], along with a suitable bound on the memory of each player.

# 3 Near-Optimal No-Regret Learning in Convex Games

In this section we describe our algorithm, *Log-Regularized Lifted Optimistic FTRL* (henceforth LRL-OFTRL). The central result of this section, Theorem 4, asserts that when all players learn using LRL-OFTRL, their regret only grows logarithmically with respect to the number of repetitions of the game. Detailed proofs for this section are available in Appendix A.

## 3.1 Setup

In the sequel, we will define and analyze the regret cumulated by LRL-OFTRL from the perspective of a generic player, omitting player subscripts.

We denote the set of strategies of the player by $\mathcal{X} \subseteq \mathbb{R}^d$. Without loss of generality, we will assume that $\mathcal{X} \subseteq [0, +\infty)^d$; otherwise, it suffices to first shift the set. Furthermore, we assume without loss of generality that there is no index $r \in [d]$ such that $\boldsymbol{x}[r] = 0$ for all $\boldsymbol{x} \in \mathcal{X}$—if not, dropping the identically-zero dimension would not alter regret. We define the *lifting of set $\mathcal{X}$* as the following set:

$$\mathbb{R}^{d+1} \supseteq \tilde{\mathcal{X}} \coloneqq \{(\lambda, \boldsymbol{y}) : \lambda \in [0, 1], \boldsymbol{y} \in \lambda \mathcal{X}\}. \tag{2}$$

Further, we define the $\ell_1$-norm $\|\mathcal{X}\|_1$ of $\mathcal{X}$ as the maximum $\ell_1$-norm of any vector $\boldsymbol{x} \in \mathcal{X}$, that is, $\|\mathcal{X}\|_1 \coloneqq \max_{\boldsymbol{x} \in \mathcal{X}} \|\boldsymbol{x}\|_1$; for example, $\|\Delta^d\|_1 = 1$.

The *logarithmic regularizer* for $\mathbb{R}^{d+1}$ is the function

$$\mathcal{R}(\lambda, \boldsymbol{y}) \coloneqq -\log \lambda - \sum_{r=1}^{d} \log \boldsymbol{y}[r], \qquad \forall (\lambda, \boldsymbol{y}) \in \mathbb{R}_{>0}^{d+1}.$$

Given any vector $(\lambda, \boldsymbol{y}) \in \tilde{\mathcal{X}} \cap \mathbb{R}_{>0}^{d+1}$, we denote with $\|\cdot\|_{(\lambda,\boldsymbol{y})}$ and $\|\cdot\|_{*,(\lambda,\boldsymbol{y})}$ the *local norms centered at* $(\lambda, \boldsymbol{y})$ induced by $\mathcal{R}(\lambda, \boldsymbol{y})$, defined as

$$\left\| \begin{pmatrix} a \\ \boldsymbol{z} \end{pmatrix} \right\|_{(\lambda,\boldsymbol{y})} \coloneqq \sqrt{\left(\frac{a}{\lambda}\right)^2 + \sum_{r=1}^{d} \left(\frac{\boldsymbol{z}[r]}{\boldsymbol{y}[r]}\right)^2}, \qquad \left\| \begin{pmatrix} a \\ \boldsymbol{z} \end{pmatrix} \right\|_{*,(\lambda,\boldsymbol{y})} \coloneqq \sqrt{(a\lambda)^2 + \sum_{r=1}^{d} (\boldsymbol{z}[r]\boldsymbol{y}[r])^2}$$

for any $(a, \boldsymbol{z}) \in \mathbb{R}^{d+1}$. These are the norms induced by the Hessian matrix of $\mathcal{R}$ at $(\lambda, \boldsymbol{y})$ and its inverse. It is a well-known fact that $\|\cdot\|_{*,(\lambda,\boldsymbol{y})}$ is the dual norm of $\|\cdot\|_{(\lambda,\boldsymbol{y})}$, and *vice versa*.

## 3.2 Overview of Our Algorithm

Our algorithm (Algorithm 1) leverages *optimistic follow the regularized leader* (OFTRL), a simple variant of FTRL introduced by Syrgkanis et al. [2015], but with some important twists. First, the optimization is performed over the lifting $\tilde{\mathcal{X}}$ of the set $\mathcal{X}$. More precisely, at every iteration the

observed utility $\boldsymbol{u}^{(t)} \in \mathbb{R}^d$ will be transformed to $\tilde{\boldsymbol{u}}^{(t)} \in \mathbb{R}^{d+1}$ according to Line 6; this ensures that $\tilde{\boldsymbol{u}}^{(t)}$ is orthogonal to the vector $(1, \boldsymbol{x}^{(t)})$. Then, this utility vector $\tilde{\boldsymbol{u}}^{(t)}$ is given as input to a regret minimizer operating over $\tilde{\mathcal{X}}$, employing OFTRL under the logarithmic regularizer $\mathcal{R}(\lambda, \boldsymbol{y})$; this step is described in Line 3. We discuss how such an optimization problem can be solved efficiently in Section 3.5. Below we point out that Line 3 is indeed well-defined.

**Proposition 1.** *For any $\eta \geq 0$ and at all times $t \in \mathbb{N}$, the* OFTRL *optimization problem on Line 3 of Algorithm 1 admits a unique optimal solution $(\lambda^{(t)}, \boldsymbol{y}^{(t)}) \in \tilde{\mathcal{X}} \cap \mathbb{R}_{>0}^{d+1}$.*

Finally, given the iterate $(\lambda^{(t)}, \boldsymbol{y}^{(t)})$ output by the OFTRL step at time $t$, our regret minimizer over $\mathcal{X}$ selects the next strategy $\boldsymbol{x}^{(t)} := \boldsymbol{y}^{(t)}/\lambda^{(t)}$ (Line 4); this is indeed a valid strategy in $\mathcal{X}$ by definition of $\tilde{\mathcal{X}}$ in (2), as well as the fact that $\lambda^{(t)} > 0$ as asserted in Proposition 1.

---

**Algorithm 1:** Log-Regularized Lifted Optimistic FTRL (LRL-OFTRL)

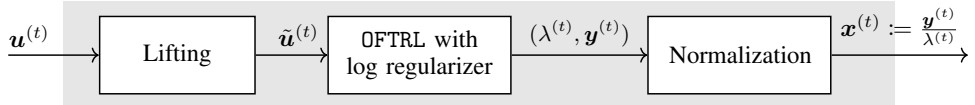

**Data:** Learning rate $\eta$

1   Set $\tilde{\boldsymbol{U}}^{(1)}, \boldsymbol{u}^{(0)} \leftarrow \boldsymbol{0} \in \mathbb{R}^{d+1}$

2   **for** $t = 1, 2, \ldots, T$ **do**

3     Set $\begin{pmatrix} \lambda^{(t)} \\ \boldsymbol{y}^{(t)} \end{pmatrix} \leftarrow \underset{(\lambda, \boldsymbol{y}) \in \tilde{\mathcal{X}}}{\arg\max} \left\{ \eta \left\langle \tilde{\boldsymbol{U}}^{(t)} + \tilde{\boldsymbol{u}}^{(t-1)}, \begin{pmatrix} \lambda \\ \boldsymbol{y} \end{pmatrix} \right\rangle + \log \lambda + \sum_{r=1}^{d} \log \boldsymbol{y}[r] \right\}$    [▷ OFTRL]

4     Play strategy $\boldsymbol{x}^{(t)} := \dfrac{\boldsymbol{y}^{(t)}}{\lambda^{(t)}} \in \mathcal{X}$    [▷ Normalization]

5     Observe $\boldsymbol{u}^{(t)} \in \mathbb{R}^d$

6     Set $\tilde{\boldsymbol{u}}^{(t)} \leftarrow \begin{pmatrix} -\langle \boldsymbol{u}^{(t)}, \boldsymbol{x}^{(t)} \rangle \\ \boldsymbol{u}^{(t)} \end{pmatrix}$    [▷ Lifting]

7     Set $\tilde{\boldsymbol{U}}^{(t+1)} \leftarrow \tilde{\boldsymbol{U}}^{(t)} + \tilde{\boldsymbol{u}}^{(t)}$

---

### 3.3   Regret Analysis

In this section, we study the regret of LRL-OFTRL under the idealized assumption that the optimization problem on Line 3 (OFTRL step) is solved exactly at each time $t$. In Section 3.5 we will relax that assumption, and study the regret of LRL-OFTRL when the solution to Line 3 is approximated using variants of Newton's method.

To study the regret $\mathrm{Reg}^T$ of LRL-OFTRL, defined in (1), it is useful to introduce the quantity $\tilde{\mathrm{Reg}}^T$, which measures the regret incurred *by the internal* OFTRL *algorithm* (Line 3) up to a time $T \in \mathbb{N}$ in the lifted space $\tilde{\mathcal{X}}$, *i.e.*,

$$\tilde{\mathrm{Reg}}^T := \max_{(\lambda^*, \boldsymbol{y}^*) \in \tilde{\mathcal{X}}} \sum_{t=1}^{T} \left\langle \tilde{\boldsymbol{u}}^{(t)}, \begin{pmatrix} \lambda^* \\ \boldsymbol{y}^* \end{pmatrix} - \begin{pmatrix} \lambda^{(t)} \\ \boldsymbol{y}^{(t)} \end{pmatrix} \right\rangle.$$

As the following theorem clarifies, there is a strong connection between $\tilde{\mathrm{Reg}}^T$ and $\mathrm{Reg}^T$.

**Theorem 2.** *For any time $T \in \mathbb{N}$ it holds that $\tilde{\mathrm{Reg}}^T = \max\{0, \mathrm{Reg}^T\}$. In particular, it follows that $\tilde{\mathrm{Reg}}^T \geq 0$ and $\mathrm{Reg}^T \leq \tilde{\mathrm{Reg}}^T$ for any $T \in \mathbb{N}$.*

The nonnegativity of $\tilde{\mathrm{Reg}}^T$ will be a crucial property in establishing Theorem 3. Further, Theorem 2 implies that a guarantee over the lifted space can be automatically translated to a regret bound over the original space $\mathcal{X}$. Now let

$$\| \cdot \|_t := \| \cdot \|_{(\lambda^{(t)}, \boldsymbol{y}^{(t)})} \qquad \text{and} \qquad \| \cdot \|_{*,t} := \| \cdot \|_{*,(\lambda^{(t)}, \boldsymbol{y}^{(t)})} \qquad (3)$$

be the local norms centered at point $(\lambda^{(t)}, \boldsymbol{y}^{(t)})$ produced by OFTRL at time $t$ (Line 3). In the next proposition we establish a refined RVU (Regret bounded by Variation in Utilities) bound in terms of this primal-dual norm pair.

**Proposition 2** (RVU bound of OFTRL in local norms)**.** *Let* $\tilde{\mathrm{Reg}}^T$ *be the regret cumulated up to time $T$ by the internal OFTRL algorithm. If* $\|\boldsymbol{u}^{(t)}\|_\infty \|\boldsymbol{x}\|_1 \leq 1$ *at all times* $t \in [\![T]\!]$, *then for any time horizon $T \in \mathbb{N}$ and learning rate* $\eta \leq \frac{1}{50}$,

$$\tilde{\mathrm{Reg}}^T \leq 4 + \frac{(d+1)\log T}{\eta} + 5\eta \sum_{t=1}^{T}\left\|\tilde{\boldsymbol{u}}^{(t)} - \tilde{\boldsymbol{u}}^{(t-1)}\right\|_{*,t}^2 - \frac{1}{27\eta}\sum_{t=1}^{T-1}\left\|\binom{\lambda^{(t+1)}}{\boldsymbol{y}^{(t+1)}} - \binom{\lambda^{(t)}}{\boldsymbol{y}^{(t)}}\right\|_t^2.$$

(We recall that $\tilde{\boldsymbol{u}}^{(0)} := \boldsymbol{0}$.) Proposition 2 differs from prior analogous results in that the regularizer is not a *barrier* over the feasible set. Next, we show that the iterates produced by OFTRL satisfy a refined notion of stability, which we refer to as *multiplicative stability*.

**Proposition 3** (Multiplicative Stability)**.** *For any time* $t \in \mathbb{N}$ *and learning rate* $\eta \leq \frac{1}{50}$, *if* $\|\boldsymbol{u}^{(t)}\|_\infty \|\boldsymbol{x}\|_1 \leq 1$,

$$\left\|\binom{\lambda^{(t+1)}}{\boldsymbol{y}^{(t+1)}} - \binom{\lambda^{(t)}}{\boldsymbol{y}^{(t)}}\right\|_t \leq 22\eta.$$

Intuitively, this property ensures that coordinates of successive iterates will have a small multiplicative deviation. We leverage this refined notion of stability to establish the following crucial lemma.

**Lemma 1.** *For any time* $t \in \mathbb{N}$ *and learning rate* $\eta \leq \frac{1}{50}$, *if* $\|\boldsymbol{u}^{(t)}\|_\infty \|\boldsymbol{x}\|_1 \leq 1$,

$$\|\boldsymbol{x}^{(t+1)} - \boldsymbol{x}^{(t)}\|_1 \leq 4\|\mathcal{X}\|_1 \left\|\binom{\lambda^{(t+1)}}{\boldsymbol{y}^{(t+1)}} - \binom{\lambda^{(t)}}{\boldsymbol{y}^{(t)}}\right\|_t.$$

Combining this lemma with Proposition 2 allows us to obtain an RVU bound for the original space $\mathcal{X}$, with no dependencies on local norms.

**Corollary 1** (RVU bound in the original (unlifted) space)**.** *Fix any time $T \in \mathbb{N}$, and suppose that* $\|\boldsymbol{u}^{(t)}\|_\infty \leq B$ *for any* $t \in [\![T]\!]$. *If* $\eta \leq \frac{1}{256B\|\mathcal{X}\|_1}$,

$$\tilde{\mathrm{Reg}}^T \leq 6B\|\mathcal{X}\|_1 + \frac{(d+1)\log T}{\eta} + 16\eta\|\mathcal{X}\|_1^2\sum_{t=1}^{T-1}\|\boldsymbol{u}^{(t+1)} - \boldsymbol{u}^{(t)}\|_\infty^2 - \frac{1}{512\eta\|\mathcal{X}\|_1^2}\sum_{t=1}^{T-1}\|\boldsymbol{x}^{(t+1)} - \boldsymbol{x}^{(t)}\|_1^2.$$

### 3.4 Main Result

So far, in Section 3.3, we have performed the analysis from the perspective of a single player, obtaining regret bounds that apply under an arbitrary sequence of utilities. Next, we assume that all players follow our dynamics such that the variation in one's utilities is now related to the variation in the joint strategies based on the smoothness condition of the utility function, connecting the last two terms of the RVU bound. Further leveraging the nonnegativity of the regrets in the lifted space, we establish that the second-order path lengths of the dynamics up to time $T$ are bounded by $O(\log T)$:

**Theorem 3.** *Suppose that Assumption 1 holds for some parameters $B, L > 0$. If all players follow LRL-OFTRL with learning rate* $\eta \leq \min\left\{\frac{1}{256B\|\mathcal{X}\|_1}, \frac{1}{128nL\|\mathcal{X}\|_1^2}\right\}$, *where* $\|\mathcal{X}\|_1 := \max_{i \in [\![n]\!]} \|\mathcal{X}_i\|_1$, *then*

$$\sum_{i=1}^{n}\sum_{t=1}^{T-1}\|\boldsymbol{x}_i^{(t+1)} - \boldsymbol{x}_i^{(t)}\|_1^2 \leq 6144n\eta B\|\mathcal{X}\|_1^3 + 1024n(d+1)\|\mathcal{X}\|_1^2\log T. \tag{4}$$

Here we made the mild assumption that each player knows the values of $n$, $L$, $B$ and $\|\mathcal{X}\|_1$ in order to appropriately tune the learning rate; otherwise, similar guarantees are possible via a standard application of the doubling trick. It is interesting to point out that (4) holds even without the concavity condition (recall Assumption 1). We next leverage Theorem 3 to establish Theorem 1, the detailed version of which is given below.

**Theorem 4** (Detailed Version of Theorem 1). *Suppose that Assumption 1 holds for some parameters* $B, L > 0$. *If all players follow* `LRL-OFTRL` *with learning rate* $\eta = \min\left\{\frac{1}{256B\|\mathcal{X}\|_1}, \frac{1}{128nL\|\mathcal{X}\|_1^2}\right\}$, *then for any* $T \in \mathbb{N}$ *the regret* $\text{Reg}_i^T$ *of each player* $i \in [\![n]\!]$ *can be bounded as*

$$\text{Reg}_i^T \leq 12B\|\mathcal{X}\|_1 + 256(d+1)\max\left\{nL\|\mathcal{X}\|_1^2, 2B\|\mathcal{X}\|_1\right\}\log T. \tag{5}$$

*Furthermore, the algorithm can be adaptive so that if player* $i$ *is instead facing adversarial utilities, then* $\text{Reg}_i^T = O(\sqrt{T})$.

For clarity, below we cast (5) of Theorem 4 in normal-form games with utilities normalized in the range $[-1, 1]$, in which case we can take $B = 1$, $L = 1$ and $\|\mathcal{X}\|_1 = 1$.

**Corollary 2** (Normal-form Games). *Suppose that all players in a normal-form game with* $n \geq 2$ *follow* `LRL-OFTRL` *with learning rate* $\eta = \frac{1}{128n}$. *Then, for any* $T \in \mathbb{N}$ *and player* $i \in [\![n]\!]$,

$$\text{Reg}_i^T \leq 12 + 256n(d+1)\log T.$$

### 3.5 Implementation and Iteration Complexity

In this section, we discuss the implementation and iteration complexity of `LRL-OFTRL`. The main difficulty in the implementation is the computation of the solution to the strictly concave *nonsmooth* constrained optimization problem in Line 3. We start by studying how the guarantees laid out in Theorem 4 are affected when the exact solution to the `OFTRL` problem (Line 4) in Algorithm 1 is replaced with an approximation. Specifically, suppose that at all times $t$ the solution to the `OFTRL` step (Line 3) in Algorithm 1 is only *approximately* solved within tolerance $\epsilon^{(t)}$, in the sense that

$$\left\|\begin{pmatrix}\lambda^{(t)}\\\boldsymbol{y}^{(t)}\end{pmatrix} - \begin{pmatrix}\lambda_\star^{(t)}\\\boldsymbol{y}_\star^{(t)}\end{pmatrix}\right\|_{(\lambda_\star^{(t)}, \boldsymbol{y}_\star^{(t)})} \leq \epsilon^{(t)}, \tag{6}$$

where $(\lambda^{(t)}, \boldsymbol{y}^{(t)}) \in \mathbb{R}_{>0}^{d+1}$ and

$$\begin{pmatrix}\lambda_\star^{(t)}\\\boldsymbol{y}_\star^{(t)}\end{pmatrix} := \arg\max_{(\lambda, \boldsymbol{y}) \in \tilde{\mathcal{X}}}\left\{\eta\left\langle\tilde{\boldsymbol{U}}^{(t)} + \tilde{\boldsymbol{u}}^{(t-1)}, \begin{pmatrix}\lambda\\\boldsymbol{y}\end{pmatrix}\right\rangle + \log\lambda + \sum_{r=1}^{d}\log\boldsymbol{y}[r]\right\}.$$

Then, it can be proven directly from the definition of regret that the guarantees given in Corollary 1 deteriorate by an additive factor proportional to the sum of the tolerances $\sum_{t=1}^{T}\epsilon^{(t)}$. As an immediate corollary, when $\epsilon^{(t)} := \epsilon := 1/T$, the conclusion of Theorem 4 applies even when the solution to the optimization problem on Line 3 is only approximated up to $\epsilon$ tolerance. Therefore, to complete our construction, it suffices to show that it is indeed possible to efficiently compute approximate solutions to the `OFTRL` step (see Appendix A.5). In the remainder of this section, we show that this is indeed the case assuming access to two different type of oracles. It should be stressed that optimizing over a general convex set introduces several challenges not present under simplex domains, inevitably leading to an increased per-iteration complexity compared to algorithms designed specifically for normal-form games—such as OMWU.

**Proximal Oracle** First, we will assume access to a *proximal oracle* in local norm for the set $\tilde{\mathcal{X}}$, that is, access to a function that is able to compute the solution to the (positive-definite) quadratic optimization problem

$$\Pi_{\tilde{\boldsymbol{w}}}(\tilde{\boldsymbol{g}}) := \arg\min_{\tilde{\boldsymbol{x}} \in \tilde{\mathcal{X}}}\left\{\tilde{\boldsymbol{g}}^\top\tilde{\boldsymbol{x}} + \frac{1}{2}\|\tilde{\boldsymbol{x}} - \tilde{\boldsymbol{w}}\|_{\tilde{\boldsymbol{w}}}^2\right\} = \arg\min_{\tilde{\boldsymbol{x}} \in \tilde{\mathcal{X}}}\left\{\tilde{\boldsymbol{g}}^\top\tilde{\boldsymbol{x}} + \frac{1}{2}\sum_{r=1}^{d+1}\left(\frac{\tilde{\boldsymbol{x}}[r]}{\tilde{\boldsymbol{w}}[r]} - 1\right)^2\right\} \tag{7}$$

for arbitrary centers $\tilde{\boldsymbol{w}} \in \mathbb{R}_{>0}^{d+1}$ and gradients $\tilde{\boldsymbol{g}} \in \mathbb{R}^{d+1}$. For certain sets $\mathcal{X} \subseteq \mathbb{R}^d$, exact proximal oracles with polynomial complexity in the dimension $d$ can be given. In particular, we show that this is the case when $\mathcal{X}$ is the strategy set of normal-form and extensive-form games by extending the approach of Gilpin [2009, pp. 128-133], as formalized below.

**Proposition 4.** *Let* $\mathcal{X} \subseteq \mathbb{R}^d$ *be the polytope of sequence-form strategies for a player in a perfect-recall extensive-form game. Then, the local proximal oracle* $\Pi_{\tilde{\boldsymbol{w}}}(\tilde{\boldsymbol{g}})$ *defined in* (7) *can be implemented exactly in time polynomial in the dimension* $d$ *for any* $\tilde{\boldsymbol{w}} \in \mathbb{R}_{>0}^{d+1}$ *and* $\tilde{\boldsymbol{g}} \in \mathbb{R}^{d+1}$.

We provide the details and a more precise statement in Appendix B. In this context, the following guarantee employs the *proximal Newton algorithm* of Tran-Dinh et al. [2015]; see Algorithm 2.

**Theorem 5** (Proximal Newton). *Given any $\epsilon > 0$, it is possible to compute $(\lambda^{(t)}, \boldsymbol{y}^{(t)}) \in \tilde{\mathcal{X}} \cap \mathbb{R}^{d+1}_{>0}$ such that (6) holds for $\epsilon^{(t)} = \epsilon$ using $O(\log\log(1/\epsilon))$ operations and $O(\log\log(1/\epsilon))$ calls to the proximal oracle defined in Equation (7).*

**Linear Maximization Oracle** Moreover, we consider having access to a weaker *linear maximization oracle (LMO)* for the set $\mathcal{X}$:

$$\mathcal{L}_{\mathcal{X}}(\boldsymbol{u}) \coloneqq \underset{\boldsymbol{x} \in \mathcal{X}}{\arg\max} \langle \boldsymbol{x}, \boldsymbol{u} \rangle. \tag{8}$$

Such an oracle is more realistic in many settings [Jaggi, 2013], and it is particularly natural in the context of games, where it can be thought of as a *best response* oracle. We point out that an LMO for $\mathcal{X}$ automatically implies an LMO for $\tilde{\mathcal{X}}$. The following guarantee follows readily by applying the *Frank-Wolfe (projected) Newton method* [Liu et al., 2020, Algotihms 1 and 2].

**Theorem 6** (Frank-Wolfe Newton). *Given any $\epsilon > 0$, it is possible to compute $(\lambda^{(t)}, \boldsymbol{y}^{(t)}) \in \tilde{\mathcal{X}} \cap \mathbb{R}^{d+1}_{>0}$ such that (6) holds for $\epsilon^{(t)} = \epsilon$ using $O(\text{poly}(1/\epsilon))$ operations and $O(\text{poly}(1/\epsilon))$ calls to the LMO oracle defined in Equation (8).*

**Experiments** Finally, while our main contribution is of theoretical nature, we also support our theory by conducting experiments on some standard extensive-form games (Appendix C). The experiments verify that under `LRL-OFTRL` the regret of each player grows as $O(\log T)$.

## 4 Conclusions

In this paper we developed `LRL-OFTRL`, a novel no-regret learning algorithm. We showed that when all players in a general convex game employ `LRL-OFTRL`, the regret of each player grows only as $O(\log T)$, thereby significantly extending and strengthening the scope of all prior work. Further, our uncoupled no-regret learning dynamics can be efficiently implemented using, for example, a proximal oracle for the underlying feasible set.

One caveat of our framework applied to the special case of normal-form games is that the dependence on the dimension is linear (Corollary 2) as opposed to logarithmic [Daskalakis et al., 2021]. Whether the entropic regularizer—which induces OMWU—can be incorporated into our framework is an important open question. Another interesting avenue for future research would be to explore having access to different types of oracles. For example, is it possible to extend Theorem 4 using a *separation oracle* for the underlying set of strategies? If so, the ellipsoid algorithm [Bubeck, 2015] would be the obvious candidate en route to implementing `LRL-OFTRL`.

## Acknowledgments

We are grateful to anonymous NeurIPS reviewers for many helpful comments. This material is based on work supported by the National Science Foundation under grants IIS-1718457, IIS-1901403, IIS-1943607, and CCF-1733556, and the ARO under award W911NF2010081. Christian Kroer is supported by the Office of Naval Research Young Investigator Program under grant N00014-22-1-2530.

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
