# A  Omitted Proofs

In this section we include all of the proofs omitted from the main body. For the convenience of the reader, we will restate each claim before proceeding with its proof.

## A.1  Preliminary Proofs

We commence with the proof of Proposition 1.

**Proposition 1.** *For any $\eta \geq 0$ and at all times $t \in \mathbb{N}$, the* OFTRL *optimization problem on Line 3 of Algorithm 1 admits a unique optimal solution* $(\lambda^{(t)}, \boldsymbol{y}^{(t)}) \in \tilde{\mathcal{X}} \cap \mathbb{R}_{>0}^{d+1}$.

*Proof.* Uniqueness follow immediately from strict convexity. In the rest of the proof we focus on the existence part.

We start by showing that there exists a point $\tilde{x} \in \tilde{\mathcal{X}}$ whose coordinates are all strictly positive. By hypothesis (see Section 3.1), for every coordinate $r \in [\![d]\!]$, there exists a point $\boldsymbol{x}_r$ such that $\boldsymbol{x}_r[r] > 0$. Hence, by convexity of $\mathcal{X} \subseteq [0, +\infty)^d$ and by definition of $\tilde{\mathcal{X}}$, the point

$$(1, \boldsymbol{x}^\circ) := \left( 1, \ \frac{1}{d} \sum_{r=1}^{d} \boldsymbol{x}_r \right).$$

is such that $(1, \boldsymbol{x}^\circ) \in \tilde{\mathcal{X}} \cap \mathbb{R}_{>0}^d$.

Let now $M$ be the $\ell_\infty$ norm of the linear part in the OFTRL step (Line 3 of Algorithm 1). Then, a *lower bound* on the optimal value $v^\star$ of objective is obtained by plugging in the point $(1, \boldsymbol{x}^\circ)$ at least

$$v^\star \geq -M(1 + \|\mathcal{X}\|_1) + \sum_{r=1}^{d} \log \boldsymbol{x}^\circ[r]. \tag{9}$$

Let now

$$m := \exp\left\{ -(2M + d)(1 + \|\mathcal{X}\|_1) + \sum_{r=1}^{d} \log \boldsymbol{x}^\circ[r] \right\} > 0. \tag{10}$$

We will show that any point $(\lambda, \boldsymbol{y}) \notin [m, +\infty) \cap \tilde{\mathcal{X}}$ cannot be optimal for the OFTRL objective. Indeed, take a point $(\lambda, \boldsymbol{y}) \notin [m, +\infty) \cap \tilde{\mathcal{X}}$. Then, at least one coordinate of $(\lambda, \boldsymbol{y})$ is strictly less than $m$. If $\lambda < m$, then the objective value at $(\lambda, \boldsymbol{y})$ is at most

$$
\begin{aligned}
M\lambda + M\|\mathcal{X}\|_1 + \log \lambda + \sum_{r=1}^{d} \log \boldsymbol{y}[r] &\leq M(1 + \|\mathcal{X}\|_1) + \log m + \sum_{r=1}^{d} \log \|\mathcal{X}\|_1 \\
&\leq M(1 + \|\mathcal{X}\|_1) + \log m + d(\|\mathcal{X}\|_1 - 1) \\
&< (M + d)(1 + \|\mathcal{X}\|_1) + \log m \\
&= -M(1 + \|\mathcal{X}\|_1) + \sum_{r=1}^{d} \log \boldsymbol{x}^\circ[r] && \text{(from (10))} \\
&\leq v^*, && \text{(from (9))}
\end{aligned}
$$

where the first inequality follows from upper bounding any coordinate of $\boldsymbol{y}$ with $\|\mathcal{X}\|_1$, and the second inequality follows from using the inequality $\log z \leq z - 1$, valid for all $z \in (0, +\infty)$. Similarly, if $\boldsymbol{y}[s] < m$ for some $s \in [\![d]\!]$, then we can upper bound the objective value at $(\lambda, \boldsymbol{y})$ as

$$
\begin{aligned}
M + M\|\mathcal{X}\|_1 + \log 1 + \sum_{r=1}^{d} \log \boldsymbol{y}[r] &\leq M(1 + \|\mathcal{X}\|_1) + \log m + \sum_{r=1}^{d} \log \|\mathcal{X}\|_1 \\
&\leq M(1 + \|\mathcal{X}\|_1) + (d - 1)(\|\mathcal{X}\|_1 - 1) + \log m \\
&< (M + d)(1 + \|\mathcal{X}\|_1) + \log m \leq v^*.
\end{aligned}
$$

So, in either case, we see that no optimal point can have any coordinate strictly less than $m$. Consequently, the maximizer of the OFTRL step lies in the set $\mathcal{S} := [m, +\infty)^{d+1} \cap \tilde{\mathcal{X}}$. Since both $[m, +\infty)^{d+1}$ and $\tilde{\mathcal{X}}$ are closed, and since $\tilde{\mathcal{X}}$ is bounded by hypothesis, the set $\mathcal{S}$ is compact. Furthermore, note that $\mathcal{S}$ is nonempty, as $(1, \boldsymbol{x}^\circ) \in \mathcal{S}$, as for any $s \in [\![d]\!]$

$$
\begin{aligned}
\log m &= -(2M + d)(1 + \|\mathcal{X}\|_1) + \sum_{r=1}^{d} \log \boldsymbol{x}^\circ[r] \\
&\le -(2M + d)(1 + \|\mathcal{X}\|_1) + \log \boldsymbol{x}^\circ[s] + (d-1)\log \|\mathcal{X}\|_1 \\
&\le -(2M + d)(1 + \|\mathcal{X}\|_1) + \log \boldsymbol{x}^\circ[s] + (d-1)(\|\mathcal{X}\|_1 - 1) \\
&\le \log \boldsymbol{x}^\circ[s],
\end{aligned}
$$

implying that $(1, \boldsymbol{x}^\circ) \in [m, +\infty)^{d+1}$. Since $\mathcal{S}$ is compact and nonempty and the objective function is continuous, the optimization problem attains an optimal solution on $\mathcal{S}$ by virtue of Weierstrass' theorem. $\qquad\square$

**Theorem 2.** *For any time $T \in \mathbb{N}$ it holds that $\tilde{\mathrm{Reg}}^T = \max\{0, \mathrm{Reg}^T\}$. In particular, it follows that $\tilde{\mathrm{Reg}}^T \ge 0$ and $\mathrm{Reg}^T \le \tilde{\mathrm{Reg}}^T$ for any $T \in \mathbb{N}$.*

*Proof.* First, by definition of $\tilde{\boldsymbol{u}}^{(t)}$ in Line 6, it follows that for any $t$,

$$
\left\langle \tilde{\boldsymbol{u}}^{(t)}, \begin{pmatrix} \lambda^{(t)} \\ \boldsymbol{y}^{(t)} \end{pmatrix} \right\rangle = \left\langle \tilde{\boldsymbol{u}}^{(t)}, \begin{pmatrix} 1 \\ \boldsymbol{x}^{(t)} \end{pmatrix} \right\rangle = 0.
$$

As a result, we have that $\max\{0, \mathrm{Reg}^T\}$ is equal to

$$
\max\left\{ 0, \max_{\boldsymbol{x}^* \in \mathcal{X}} \sum_{t=1}^{T} \langle \boldsymbol{u}^{(t)}, \boldsymbol{x}^* - \boldsymbol{x}^{(t)} \rangle \right\} = \max\left\{ 0, \max_{\boldsymbol{x}^* \in \mathcal{X}} \sum_{t=1}^{T} \left\langle \tilde{\boldsymbol{u}}^{(t)}, \begin{pmatrix} 1 \\ \boldsymbol{x}^* \end{pmatrix} - \begin{pmatrix} 1 \\ \boldsymbol{x}^{(t)} \end{pmatrix} \right\rangle \right\}
$$

$$
= \max\left\{ 0, \max_{\boldsymbol{x}^* \in \mathcal{X}} \sum_{t=1}^{T} \left\langle \tilde{\boldsymbol{u}}^{(t)}, \begin{pmatrix} 1 \\ \boldsymbol{x}^* \end{pmatrix} \right\rangle \right\} = \max_{(\lambda^*, \boldsymbol{y}^*) \in \tilde{\mathcal{X}}} \sum_{t=1}^{T} \left\langle \tilde{\boldsymbol{u}}^{(t)}, \begin{pmatrix} \lambda^* \\ \boldsymbol{y}^* \end{pmatrix} \right\rangle
$$

$$
= \max_{(\lambda^*, \boldsymbol{y}^*) \in \tilde{\mathcal{X}}} \sum_{t=1}^{T} \left\langle \tilde{\boldsymbol{u}}^{(t)}, \begin{pmatrix} \lambda^* \\ \boldsymbol{y}^* \end{pmatrix} - \begin{pmatrix} \lambda^{(t)} \\ \boldsymbol{y}^{(t)} \end{pmatrix} \right\rangle = \tilde{\mathrm{Reg}}^T,
$$

as we wanted to show. $\qquad\square$

## A.2 Analysis of OFTRL with Logarithmic Regularizer

For notational convenience, we define the log-regularizer $\psi : \tilde{\mathcal{X}} \to \mathbb{R}_{\ge 0}$ as

$$
\psi(\tilde{\boldsymbol{x}}) := -\frac{1}{\eta} \sum_{r=1}^{d+1} \log \tilde{\boldsymbol{x}}[r],
$$

and its induced Bregman divergence

$$
D_\psi\left(\tilde{\boldsymbol{x}} \,\|\, \tilde{\boldsymbol{z}}\right) := \frac{1}{\eta} \sum_{r=1}^{d+1} h\left(\frac{\tilde{\boldsymbol{x}}[r]}{\tilde{\boldsymbol{z}}[r]}\right), \quad \text{where } h(a) = a - 1 - \ln(a).
$$

Moreover, we define

$$
\tilde{\boldsymbol{x}}^{(t)} = \arg\max_{\tilde{\boldsymbol{x}} \in \tilde{\mathcal{X}}} -F_t(\tilde{\boldsymbol{x}}) = \arg\min_{\tilde{\boldsymbol{x}} \in \tilde{\mathcal{X}}} F_t(\tilde{\boldsymbol{x}}), \quad \text{where } F_t(\tilde{\boldsymbol{x}}) = -\left\langle \tilde{\boldsymbol{U}}^{(t)} + \tilde{\boldsymbol{u}}^{(t-1)}, \tilde{\boldsymbol{x}} \right\rangle + \psi(\tilde{\boldsymbol{x}}).
$$

$$(11)$$

We note that $F_t$ is a convex function for each $t$ and $\tilde{\boldsymbol{x}}^{(t)}$ is exactly equal to $\begin{pmatrix} \lambda^{(t)} \\ \boldsymbol{y}^{(t)} \end{pmatrix}$ computed by Algorithm 1. Further, we define an auxiliary sequence $\{\tilde{\boldsymbol{z}}^{(t)}\}_{t=1,2,\dots}$ defined as follows.

$$
\tilde{\boldsymbol{z}}^{(t)} = \arg\max_{\tilde{\boldsymbol{x}} \in \tilde{\mathcal{X}}} -G_t(\tilde{\boldsymbol{x}}) = \arg\min_{\tilde{\boldsymbol{x}} \in \tilde{\mathcal{X}}} G_t(\tilde{\boldsymbol{x}}), \quad \text{where } G_t(\tilde{\boldsymbol{x}}) = -\left\langle \tilde{\boldsymbol{U}}^{(t)}, \tilde{\boldsymbol{x}} \right\rangle + \psi(\tilde{\boldsymbol{x}}). \qquad (12)
$$

Similarly, $G_t$ is a convex function for each $t$. We also recall the primal and dual norm notation:

$$\|\tilde{z}\|_t = \sum_{r=1}^{d+1} \left( \frac{\tilde{z}[r]}{\tilde{x}^{(t)}[r]} \right)^2, \quad \|\tilde{z}\|_{*,t} = \sum_{r=1}^{d+1} \left( \tilde{x}^{(t)}[r]\tilde{z}[r] \right)^2.$$

Finally, for a $(d+1) \times (d+1)$ positive definite matrix $\mathbf{M}$, we use $\|\tilde{z}\|_{\mathbf{M}}$ to denote the induced quadratic norm $\sqrt{\tilde{z}^\top \mathbf{M}\tilde{z}}$. We are now ready to establish Proposition 2.

**Proposition 2** (RVU bound of OFTRL in local norms). *Let $\tilde{\mathrm{Reg}}^T$ be the regret cumulated up to time $T$ by the internal OFTRL algorithm. If $\|\boldsymbol{u}^{(t)}\|_\infty \|\boldsymbol{x}\|_1 \le 1$ at all times $t \in [\![T]\!]$, then for any time horizon $T \in \mathbb{N}$ and learning rate $\eta \le \frac{1}{50}$,*

$$\tilde{\mathrm{Reg}}^T \le 4 + \frac{(d+1)\log T}{\eta} + 5\eta \sum_{t=1}^{T} \left\| \tilde{\boldsymbol{u}}^{(t)} - \tilde{\boldsymbol{u}}^{(t-1)} \right\|_{*,t}^2 - \frac{1}{27\eta} \sum_{t=1}^{T-1} \left\| \begin{pmatrix} \lambda^{(t+1)} \\ \boldsymbol{y}^{(t+1)} \end{pmatrix} - \begin{pmatrix} \lambda^{(t)} \\ \boldsymbol{y}^{(t)} \end{pmatrix} \right\|_t^2.$$

*Proof of Proposition 2.* For any comparator $\tilde{\boldsymbol{x}} \in \tilde{\mathcal{X}}$, define $\tilde{\boldsymbol{x}}' = \frac{T-1}{T} \cdot \tilde{\boldsymbol{x}} + \frac{1}{T} \cdot \tilde{\boldsymbol{x}}^{(1)} \in \tilde{\mathcal{X}}$, where we recall $\tilde{\boldsymbol{x}}^{(1)} = \arg\min_{\tilde{\boldsymbol{x}} \in \tilde{\mathcal{X}}} F_1(\tilde{\boldsymbol{x}}) = \arg\min_{\tilde{\boldsymbol{x}} \in \tilde{\mathcal{X}}} \psi(\tilde{\boldsymbol{x}})$. Then, we have

$$\sum_{t=1}^{T} \left\langle \tilde{\boldsymbol{x}} - \tilde{\boldsymbol{x}}^{(t)}, \tilde{\boldsymbol{u}}^{(t)} \right\rangle = \sum_{t=1}^{T} \left\langle \tilde{\boldsymbol{x}} - \tilde{\boldsymbol{x}}', \tilde{\boldsymbol{u}}^{(t)} \right\rangle + \sum_{t=1}^{T} \left\langle \tilde{\boldsymbol{x}}' - \tilde{\boldsymbol{x}}^{(t)}, \tilde{\boldsymbol{u}}^{(t)} \right\rangle$$

$$= \frac{1}{T} \sum_{t=1}^{T} \left\langle \tilde{\boldsymbol{x}} - \tilde{\boldsymbol{x}}^{(1)}, \tilde{\boldsymbol{u}}^{(t)} \right\rangle + \sum_{t=1}^{T} \left\langle \tilde{\boldsymbol{x}}' - \tilde{\boldsymbol{x}}^{(t)}, \tilde{\boldsymbol{u}}^{(t)} \right\rangle$$

$$\le 4 + \sum_{t=1}^{T} \left\langle \tilde{\boldsymbol{x}}' - \tilde{\boldsymbol{x}}^{(t)}, \tilde{\boldsymbol{u}}^{(t)} \right\rangle,$$

where the last inequality follows from Cauchy-Schwarz together with the assumption that $\|\boldsymbol{u}^{(t)}\|_\infty \le \frac{1}{\|\mathcal{X}\|_1}$.

Now, by standard Optimistic FTRL analysis (see Lemma 2), the last term $\sum_{t=1}^{T} \left\langle \tilde{\boldsymbol{x}}' - \tilde{\boldsymbol{x}}^{(t)}, \tilde{\boldsymbol{u}}^{(t)} \right\rangle$ (cumulative regret against $\tilde{\boldsymbol{x}}'$) is bounded by

$$\sum_{t=1}^{T} \left\langle \tilde{\boldsymbol{x}}' - \tilde{\boldsymbol{x}}^{(t)}, \tilde{\boldsymbol{u}}^{(t)} \right\rangle \le \psi(\tilde{\boldsymbol{x}}') - \psi(\tilde{\boldsymbol{x}}^{(1)}) + \sum_{t=1}^{T} \left\langle \tilde{\boldsymbol{z}}^{(t+1)} - \tilde{\boldsymbol{x}}^{(t)}, \tilde{\boldsymbol{u}}^{(t)} - \tilde{\boldsymbol{u}}^{(t-1)} \right\rangle$$

$$- \sum_{t=1}^{T} \left( D_\psi\left( \tilde{\boldsymbol{x}}^{(t)} \,\big\|\, \tilde{\boldsymbol{z}}^{(t)} \right) + D_\psi\left( \tilde{\boldsymbol{z}}^{(t+1)} \,\big\|\, \tilde{\boldsymbol{x}}^{(t)} \right) \right).$$

For the term $\psi(\tilde{\boldsymbol{x}}') - \psi(\tilde{\boldsymbol{x}}^{(1)})$, a direct calculation using definitions shows

$$\psi(\tilde{\boldsymbol{x}}') - \psi(\tilde{\boldsymbol{x}}^{(1)}) = \frac{1}{\eta} \sum_{i=1}^{d+1} \log \frac{\tilde{\boldsymbol{x}}^{(1)}[i]}{\tilde{\boldsymbol{x}}'[i]} \le \frac{d+1}{\eta} \log T.$$

For the other terms, we apply Lemma 3 and Lemma 5, which completes the proof. $\qquad\square$

**Lemma 2.** *The update rule* (11) *ensures the following for any $\tilde{\boldsymbol{x}} \in \tilde{\mathcal{X}}$:*

$$\sum_{t=1}^{T} \left\langle \tilde{\boldsymbol{x}} - \tilde{\boldsymbol{x}}^{(t)}, \tilde{\boldsymbol{u}}^{(t)} \right\rangle \le \psi(\tilde{\boldsymbol{x}}) - \psi(\tilde{\boldsymbol{x}}^{(1)}) + \sum_{t=1}^{T} \left\langle \tilde{\boldsymbol{z}}^{(t+1)} - \tilde{\boldsymbol{x}}^{(t)}, \tilde{\boldsymbol{u}}^{(t)} - \tilde{\boldsymbol{u}}^{(t-1)} \right\rangle$$

$$- \sum_{t=1}^{T} \left( D_\psi\left( \tilde{\boldsymbol{x}}^{(t)} \,\big\|\, \tilde{\boldsymbol{z}}^{(t)} \right) + D_\psi\left( \tilde{\boldsymbol{z}}^{(t+1)} \,\big\|\, \tilde{\boldsymbol{x}}^{(t)} \right) \right).$$

*Proof.* First note that for any convex function $F : \tilde{\mathcal{X}} \to \mathbb{R}$ and a minimizer $\tilde{\boldsymbol{x}}^\star$, we have for any $\tilde{\boldsymbol{x}} \in \tilde{\mathcal{X}}$:

$$F(\tilde{\boldsymbol{x}}^\star) = F(\tilde{\boldsymbol{x}}) - \langle \nabla F(\tilde{\boldsymbol{x}}^\star), \tilde{\boldsymbol{x}} - \tilde{\boldsymbol{x}}^\star \rangle - D_F(\tilde{\boldsymbol{x}}, \tilde{\boldsymbol{x}}^\star) \le F(\tilde{\boldsymbol{x}}) - D_F(\tilde{\boldsymbol{x}}, \tilde{\boldsymbol{x}}^\star),$$

where $D_F$ is the Bregman Divergence induced by $F$ and the inequality is by the first-order optimality. Using this fact and the optimality of $\tilde{z}^{(t)}$, we have

$$G_t(\tilde{z}^{(t)}) \leq G_t(\tilde{x}^{(t)}) - D_\psi\left(\tilde{x}^{(t)} \,\big\|\, \tilde{z}^{(t)}\right)$$

$$= F_t(\tilde{x}^{(t)}) + \left\langle \tilde{x}^{(t)}, \tilde{u}^{(t-1)}\right\rangle - D_\psi\left(\tilde{x}^{(t)} \,\big\|\, \tilde{z}^{(t)}\right)$$

Similarly, using the optimality of $\tilde{x}^{(t)}$, we have

$$F_t(\tilde{x}^{(t)}) \leq F_t(\tilde{z}^{(t+1)}) - D_\psi\left(\tilde{z}^{(t+1)} \,\big\|\, \tilde{x}^{(t)}\right)$$

$$= G_{t+1}(\tilde{z}^{(t+1)}) + \left\langle \tilde{z}^{(t+1)}, \tilde{u}^{(t)} - \tilde{u}^{(t-1)}\right\rangle - D_\psi\left(\tilde{z}^{(t+1)} \,\big\|\, \tilde{x}^{(t)}\right)$$

Combining the inequalities and summing over $t$, we have

$$G_1(\tilde{z}^{(1)}) \leq G_{T+1}(\tilde{z}^{(T+1)}) + \sum_{t=1}^{T}\left(\left\langle \tilde{x}^{(t)}, \tilde{u}^{(t)}\right\rangle + \left\langle \tilde{z}^{(t+1)} - \tilde{x}^{(t)}, \tilde{u}^{(t)} - \tilde{u}^{(t-1)}\right\rangle\right)$$

$$+ \sum_{t=1}^{T}\left(-D_\psi\left(\tilde{x}^{(t)} \,\big\|\, \tilde{z}^{(t)}\right) - D_\psi\left(\tilde{z}^{(t+1)} \,\big\|\, \tilde{x}^{(t)}\right)\right).$$

Observe that $G_1(\tilde{z}^{(1)}) = \psi(\tilde{x}^{(1)})$ and $G_{T+1}(\tilde{z}^{(T+1)}) \leq -\left\langle \tilde{x}, \tilde{U}^{(T+1)}\right\rangle + \psi(\tilde{x})$. Rearranging then proves the lemma. $\qquad\square$

**Lemma 3.** *If $\eta \leq \frac{1}{50}$, then we have*

$$\left\|\tilde{z}^{(t+1)} - \tilde{x}^{(t)}\right\|_t \leq 5\eta\left\|\tilde{u}^{(t)} - \tilde{u}^{(t-1)}\right\|_{*,t} \leq 10\sqrt{2}\eta \leq 15\eta, \tag{13}$$

$$\left\|\tilde{x}^{(t+1)} - \tilde{x}^{(t)}\right\|_t \leq 5\eta\left\|2\tilde{u}^{(t)} - \tilde{u}^{(t-1)}\right\|_{*,t} \leq 15\sqrt{2}\eta \leq 22\eta. \tag{14}$$

*Proof.* The second part of both inequalities is clear by definitions:

$$\left\|\tilde{u}^{(t)} - \tilde{u}^{(t-1)}\right\|_{*,t}^2 = \left(\lambda^{(t)}\left(\left\langle x^{(t)}, u^{(t)}\right\rangle - \left\langle x^{(t-1)}, u^{(t-1)}\right\rangle\right)\right)^2 + \sum_{r=1}^{d}\left(y^{(t)}[r]\left(u^{(t)}[r] - u^{(t-1)}[r]\right)\right)^2$$

$$\leq 4(\lambda^{(t)})^2 + \frac{4}{\|\mathcal{X}\|_1^2}\sum_{r=1}^{d}\left(y^{(t)}[r]\right)^2 \leq 8,$$

where we use $\left\langle x^{(\tau)}, u^{(\tau)}\right\rangle \leq \|x^{(\tau)}\|_1\|u^{(\tau)}\|_\infty \leq 1$ and $|u^{(\tau)}[r]| \leq \frac{1}{\|\mathcal{X}\|_1}$ for any time $\tau$ and any coordinate $r$ by the assumption, and similarly,

$$\left\|2\tilde{u}^{(t)} - \tilde{u}^{(t-1)}\right\|_{*,t}^2 = \left(\lambda^{(t)}\left(2\left\langle x^{(t)}, u^{(t)}\right\rangle - \left\langle x^{(t-1)}, u^{(t-1)}\right\rangle\right)\right)^2 + \sum_{r=1}^{d}\left(y^{(t)}[r]\left(2u^{(t)}[r] - u^{(t-1)}[r]\right)\right)^2$$

$$\leq 9(\lambda^{(t)})^2 + \frac{9}{\|\mathcal{X}\|_1^2}\sum_{r=1}^{d}\left(y^{(t)}[r]\right)^2 \leq 18$$

To prove the first inequality in Eq. (13), let $\mathcal{E}_t = \left\{\tilde{x} : \left\|\tilde{x} - \tilde{x}^{(t)}\right\|_t \leq 5\eta\left\|\tilde{u}^{(t)} - \tilde{u}^{(t-1)}\right\|_{*,t}\right\}$. Noticing that $\tilde{z}^{(t+1)}$ is the minimizer of the convex function $G_{t+1}$, to show $\tilde{z}^{(t+1)} \in \mathcal{E}_t$, it suffices to show that for all $\tilde{x}$ on the boundary of $\mathcal{E}_t$, we have $G_{t+1}(\tilde{x}) \geq G_{t+1}(\tilde{x}^{(t)})$. Indeed, using

Taylor's theorem, for any such $\tilde{\boldsymbol{x}}$, there is a point $\boldsymbol{\xi}$ on the line segment between $\tilde{\boldsymbol{x}}^{(t)}$ and $\tilde{\boldsymbol{x}}$ such that

$$
\begin{aligned}
G_{t+1}(\tilde{\boldsymbol{x}}) &= G_{t+1}(\tilde{\boldsymbol{x}}^{(t)}) + \left\langle \nabla G_{t+1}(\tilde{\boldsymbol{x}}^{(t)}), \tilde{\boldsymbol{x}} - \tilde{\boldsymbol{x}}^{(t)} \right\rangle + \frac{1}{2} \left\| \tilde{\boldsymbol{x}} - \tilde{\boldsymbol{x}}^{(t)} \right\|_{\nabla^2 G_{t+1}(\boldsymbol{\xi})}^2 \\
&= G_{t+1}(\tilde{\boldsymbol{x}}^{(t)}) - \left\langle \tilde{\boldsymbol{u}}^{(t)} - \tilde{\boldsymbol{u}}^{(t-1)}, \tilde{\boldsymbol{x}} - \tilde{\boldsymbol{x}}^{(t)} \right\rangle + \left\langle \nabla F_t(\tilde{\boldsymbol{x}}^{(t)}), \tilde{\boldsymbol{x}} - \tilde{\boldsymbol{x}}^{(t)} \right\rangle + \frac{1}{2} \left\| \tilde{\boldsymbol{x}} - \tilde{\boldsymbol{x}}^{(t)} \right\|_{\nabla^2 \psi(\boldsymbol{\xi})}^2 \\
&\geq G_{t+1}(\tilde{\boldsymbol{x}}^{(t)}) - \left\langle \tilde{\boldsymbol{u}}^{(t)} - \tilde{\boldsymbol{u}}^{(t-1)}, \tilde{\boldsymbol{x}} - \tilde{\boldsymbol{x}}^{(t)} \right\rangle + \frac{1}{2} \left\| \tilde{\boldsymbol{x}} - \tilde{\boldsymbol{x}}^{(t)} \right\|_{\nabla^2 \psi(\boldsymbol{\xi})}^2 \\
&\qquad\qquad\qquad\qquad\qquad\qquad\qquad\qquad\qquad \text{(by the optimality of } \tilde{\boldsymbol{x}}^{(t)}) \\
&\geq G_{t+1}(\tilde{\boldsymbol{x}}^{(t)}) - \left\| \tilde{\boldsymbol{u}}^{(t)} - \tilde{\boldsymbol{u}}^{(t-1)} \right\|_{*,t} \left\| \tilde{\boldsymbol{x}} - \tilde{\boldsymbol{x}}^{(t)} \right\|_t + \frac{1}{2} \left\| \tilde{\boldsymbol{x}} - \tilde{\boldsymbol{x}}^{(t)} \right\|_{\nabla^2 \psi(\boldsymbol{\xi})}^2 . \\
&\qquad\qquad\qquad\qquad\qquad\qquad\qquad\qquad\qquad \text{(by Hölder's inequality)} \\
&\geq G_{t+1}(\tilde{\boldsymbol{x}}^{(t)}) - \left\| \tilde{\boldsymbol{u}}^{(t)} - \tilde{\boldsymbol{u}}^{(t-1)} \right\|_{*,t} \left\| \tilde{\boldsymbol{x}} - \tilde{\boldsymbol{x}}^{(t)} \right\|_t + \frac{2}{9\eta} \left\| \tilde{\boldsymbol{x}} - \tilde{\boldsymbol{x}}^{(t)} \right\|_t^2 \qquad\qquad (\star) \\
&= G_{t+1}(\tilde{\boldsymbol{x}}^{(t)}) + \frac{5}{9}\eta \left\| \tilde{\boldsymbol{u}}^{(t)} - \tilde{\boldsymbol{u}}^{(t-1)} \right\|_{*,t}^2 \qquad (\left\| \tilde{\boldsymbol{x}} - \tilde{\boldsymbol{x}}^{(t)} \right\|_t = 5\eta \left\| \tilde{\boldsymbol{u}}^{(t)} - \tilde{\boldsymbol{u}}^{(t-1)} \right\|_{*,t}) \\
&\geq G_{t+1}(\tilde{\boldsymbol{x}}^{(t)}).
\end{aligned}
$$

Here, the inequality $(\star)$ holds because Lemma 4 (together with the condition $\eta \leq \frac{1}{50}$) shows $\frac{1}{2}\tilde{\boldsymbol{x}}^{(t)}[i] \leq \tilde{\boldsymbol{x}}[i] \leq \frac{3}{2}\tilde{\boldsymbol{x}}^{(t)}[i]$, which implies $\frac{1}{2}\tilde{\boldsymbol{x}}^{(t)}[i] \leq \boldsymbol{\xi}[i] \leq \frac{3}{2}\tilde{\boldsymbol{x}}^{(t)}[i]$ as well, and thus $\nabla^2 \psi(\boldsymbol{\xi}) \succeq \frac{4}{9}\nabla^2 \psi(\tilde{\boldsymbol{x}}^{(t)})$. This finishes the proof for Eq. (13). The first inequality of Eq. (14) can be proven in the same manner. $\qquad\square$

**Proposition 3** (Multiplicative Stability). *For any time $t \in \mathbb{N}$ and learning rate $\eta \leq \frac{1}{50}$, if $\|\boldsymbol{u}^{(t)}\|_\infty \|\boldsymbol{x}\|_1 \leq 1$,*

$$
\left\| \begin{pmatrix} \lambda^{(t+1)} \\ \boldsymbol{y}^{(t+1)} \end{pmatrix} - \begin{pmatrix} \lambda^{(t)} \\ \boldsymbol{y}^{(t)} \end{pmatrix} \right\|_t \leq 22\eta.
$$

*Proof.* The statement is proved in Lemma 3. $\qquad\square$

**Lemma 4.** *If $\tilde{\boldsymbol{x}}$ satisfies $\|\tilde{\boldsymbol{x}} - \tilde{\boldsymbol{x}}^{(t)}\|_t \leq \frac{1}{2}$, then $\frac{1}{2}\tilde{\boldsymbol{x}}^{(t)}[i] \leq \tilde{\boldsymbol{x}}[i] \leq \frac{3}{2}\tilde{\boldsymbol{x}}^{(t)}[i]$ for every coordinate $i$.*

*Proof.* By definition, $\|\tilde{\boldsymbol{x}} - \tilde{\boldsymbol{x}}^{(t)}\|_t \leq \frac{1}{2}$ implies for any $i$, $\frac{|\tilde{\boldsymbol{x}}[i] - \tilde{\boldsymbol{x}}^{(t)}[i]|}{\tilde{\boldsymbol{x}}^{(t)}[i]} \leq \frac{1}{2}$, and thus $\frac{1}{2}\tilde{\boldsymbol{x}}^{(t)}[i] \leq \tilde{\boldsymbol{x}}[i] \leq \frac{3}{2}\tilde{\boldsymbol{x}}^{(t)}[i]$. $\qquad\square$

**Lemma 5.** *If $\eta \leq \frac{1}{50}$, then we have*

$$
\sum_{t=1}^{T} \left( D_\psi \left( \tilde{\boldsymbol{x}}^{(t)} \,\Big\|\, \tilde{\boldsymbol{z}}^{(t)} \right) + D_\psi \left( \tilde{\boldsymbol{z}}^{(t+1)} \,\Big\|\, \tilde{\boldsymbol{x}}^{(t)} \right) \right) \geq \frac{1}{27\eta} \sum_{t=1}^{T-1} \left\| \tilde{\boldsymbol{x}}^{(t+1)} - \tilde{\boldsymbol{x}}^{(t)} \right\|_t^2 .
$$

*Proof.* Recall $h(a) = a - 1 - \ln(a)$ and $D_\psi\left(\tilde{\boldsymbol{x}} \,\|\, \tilde{\boldsymbol{z}}\right) = \frac{1}{\eta} \sum_{i=1}^{d+1} h\left(\frac{\tilde{\boldsymbol{x}}[i]}{\tilde{\boldsymbol{z}}[i]}\right)$. We proceed as

$$\sum_{t=1}^{T} \left( D_\psi\left(\tilde{\boldsymbol{x}}^{(t)} \,\middle\|\, \tilde{\boldsymbol{z}}^{(t)}\right) + D_\psi\left(\tilde{\boldsymbol{z}}^{(t+1)} \,\middle\|\, \tilde{\boldsymbol{x}}^{(t)}\right) \right)$$

$$\geq \sum_{t=1}^{T-1} \left( D_\psi\left(\tilde{\boldsymbol{x}}^{(t+1)} \,\middle\|\, \tilde{\boldsymbol{z}}^{(t+1)}\right) + D_\psi\left(\tilde{\boldsymbol{z}}^{(t+1)} \,\middle\|\, \tilde{\boldsymbol{x}}^{(t)}\right) \right)$$

$$= \frac{1}{\eta} \sum_{t=1}^{T-1} \sum_{i=1}^{d+1} \left( h\left(\frac{\tilde{\boldsymbol{x}}^{(t+1)}[i]}{\tilde{\boldsymbol{z}}^{(t+1)}[i]}\right) + h\left(\frac{\tilde{\boldsymbol{z}}^{(t+1)}[i]}{\tilde{\boldsymbol{x}}^{(t)}[i]}\right) \right)$$

$$\geq \frac{1}{6\eta} \sum_{t=1}^{T-1} \sum_{i=1}^{d+1} \left( \frac{(\tilde{\boldsymbol{x}}^{(t+1)}[i] - \tilde{\boldsymbol{z}}^{(t+1)}[i])^2}{\left(\tilde{\boldsymbol{z}}^{(t+1)}[i]\right)^2} + \frac{(\tilde{\boldsymbol{z}}^{(t+1)}[i] - \tilde{\boldsymbol{x}}^{(t)}[i])^2}{(\tilde{\boldsymbol{x}}^{(t)}[i])^2} \right)$$

$$\left( h(y) \geq \tfrac{(y-1)^2}{6} \text{ for } y \in [\tfrac{1}{3}, 3] \right)$$

$$\geq \frac{2}{27\eta} \sum_{t=1}^{T-1} \sum_{i=1}^{d+1} \left( \frac{(\tilde{\boldsymbol{x}}^{(t+1)}[i] - \tilde{\boldsymbol{z}}^{(t+1)}[i])^2}{\left(\tilde{\boldsymbol{x}}^{(t)}[i]\right)^2} + \frac{(\tilde{\boldsymbol{z}}^{(t+1)}[i] - \tilde{\boldsymbol{x}}^{(t)}[i])^2}{(\tilde{\boldsymbol{x}}^{(t)}[i])^2} \right)$$

$$\geq \frac{1}{27\eta} \sum_{t=1}^{T-1} \sum_{i=1}^{d+1} \left( \frac{(\tilde{\boldsymbol{x}}^{(t+1)}[i] - \tilde{\boldsymbol{x}}^{(t)}[i])^2}{\left(\tilde{\boldsymbol{x}}^{(t)}[i]\right)^2} \right) = \frac{1}{27\eta} \sum_{t=1}^{T-1} \left\| \tilde{\boldsymbol{x}}^{(t+1)} - \tilde{\boldsymbol{x}}^{(t)} \right\|_t^2 .$$

Here, the second and the third inequality hold because by Lemma 3 and Lemma 4, we have $\frac{1}{2} \leq \frac{\tilde{\boldsymbol{z}}^{(t+1)}[i]}{\tilde{\boldsymbol{x}}^{(t)}[i]} \leq \frac{3}{2}$ and $\frac{1}{2} \leq \frac{\tilde{\boldsymbol{x}}^{(t+1)}[i]}{\tilde{\boldsymbol{x}}^{(t)}[i]} \leq \frac{3}{2}$, and thus $\frac{1}{3} \leq \frac{\tilde{\boldsymbol{x}}^{(t+1)}[i]}{\tilde{\boldsymbol{z}}^{(t+1)}[i]} \leq 3$. □

### A.3 RVU Bound in the Original Space

Next, we establish an RVU bound in the original (unlifted) space, namely Corollary 1. To this end, we first proceed with the proof of Lemma 1, which boils down to the following simple claim.

**Lemma 6.** *Let $(\lambda, \boldsymbol{y}), (\lambda', \boldsymbol{y}') \in \tilde{\mathcal{X}} \cap \mathbb{R}_{>0}^{d+1}$ be arbitrary points such that*

$$\left\| \begin{pmatrix} \lambda' \\ \boldsymbol{y}' \end{pmatrix} - \begin{pmatrix} \lambda \\ \boldsymbol{y} \end{pmatrix} \right\|_{(\lambda, \boldsymbol{y})} \leq \frac{1}{2}.$$

*Then,*

$$\left\| \frac{\boldsymbol{y}}{\lambda} - \frac{\boldsymbol{y}'}{\lambda'} \right\|_1 \leq 4 \|\mathcal{X}\|_1 \cdot \left\| \begin{pmatrix} \lambda' \\ \boldsymbol{y}' \end{pmatrix} - \begin{pmatrix} \lambda \\ \boldsymbol{y} \end{pmatrix} \right\|_{(\lambda, \boldsymbol{y})}.$$

*Proof.* Let $\mu$ be defined as

$$\mu := \max\left\{ \left| \frac{\lambda'}{\lambda} - 1 \right|, \max_{r \in \llbracket d \rrbracket} \left| \frac{\boldsymbol{y}'[r]}{\boldsymbol{y}[r]} - 1 \right| \right\}. \tag{15}$$

By definition,

$$\left| \frac{\lambda'}{\lambda} - 1 \right| \leq \mu,$$

which in turn implies that

$$(1 - \mu)\lambda \leq \lambda' \leq (1 + \mu)\lambda. \tag{16}$$

Similarly, for any $r \in \llbracket d \rrbracket$,

$$(1 - \mu)\boldsymbol{y}[r] \leq \boldsymbol{y}'[r] \leq (1 + \mu)\boldsymbol{y}[r]. \tag{17}$$

As a result, combining (16) and (17) we get that for any $r \in \llbracket d \rrbracket$,

$$\frac{\boldsymbol{y}'[r]}{\lambda'} - \frac{\boldsymbol{y}[r]}{\lambda} \leq \left( \frac{1 + \mu}{1 - \mu} - 1 \right) \frac{\boldsymbol{y}[r]}{\lambda} \leq 4\mu \frac{\boldsymbol{y}[r]}{\lambda} = 4\mu \boldsymbol{x}[r],$$

since $\mu \leq \frac{1}{2}$. Similarly, by (16) and (17),

$$\frac{\boldsymbol{y}[r]}{\lambda} - \frac{\boldsymbol{y}'[r]}{\lambda'} \leq \left(1 - \frac{1-\mu}{1+\mu}\right) \frac{\boldsymbol{y}[r]}{\lambda} \leq 2\mu \frac{\boldsymbol{y}[r]}{\lambda} = 2\mu \boldsymbol{x}[r].$$

Thus, it follows that

$$\left|\frac{\boldsymbol{y}'[r]}{\lambda'} - \frac{\boldsymbol{y}[r]}{\lambda}\right| \leq 4\mu \boldsymbol{x}[r],$$

in turn implying that

$$\|\boldsymbol{x}' - \boldsymbol{x}\|_1 = \sum_{r=1}^{d} \left|\frac{\boldsymbol{y}'[r]}{\lambda'} - \frac{\boldsymbol{y}[r]}{\lambda}\right| \leq 4\mu \sum_{r=1}^{d} \boldsymbol{x}[r] \leq 4\mu \|\mathcal{X}\|_1. \tag{18}$$

Moreover, by definition of (15),

$$(\mu)^2 \leq \left\|\begin{pmatrix}\lambda'\\\boldsymbol{y}'\end{pmatrix} - \begin{pmatrix}\lambda\\\boldsymbol{y}\end{pmatrix}\right\|_t^2.$$

Finally, combining this bound with (18) concludes the proof. $\qquad\square$

**Lemma 1.** *For any time $t \in \mathbb{N}$ and learning rate $\eta \leq \frac{1}{50}$, if $\|\boldsymbol{u}^{(t)}\|_\infty \|\boldsymbol{x}\|_1 \leq 1$,*

$$\|\boldsymbol{x}^{(t+1)} - \boldsymbol{x}^{(t)}\|_1 \leq 4\|\mathcal{X}\|_1 \left\|\begin{pmatrix}\lambda^{(t+1)}\\\boldsymbol{y}^{(t+1)}\end{pmatrix} - \begin{pmatrix}\lambda^{(t)}\\\boldsymbol{y}^{(t)}\end{pmatrix}\right\|_t.$$

*Proof.* Since $\eta \leq \frac{1}{50}$ by assumption, we have

$$\left\|\boldsymbol{x}^{(t+1)} - \boldsymbol{x}^{(t)}\right\|_t \leq 22\eta < \frac{1}{2}.$$

Hence, we are in the domain of applicability of Lemma 6, which immediately yields the statement. $\qquad\square$

**Corollary 1** (RVU bound in the original (unlifted) space)**.** *Fix any time $T \in \mathbb{N}$, and suppose that $\|\boldsymbol{u}^{(t)}\|_\infty \leq B$ for any $t \in [\![T]\!]$. If $\eta \leq \frac{1}{256B\|\mathcal{X}\|_1}$,*

$$\tilde{\mathrm{Reg}}^T \leq 6B\|\mathcal{X}\|_1 + \frac{(d+1)\log T}{\eta} + 16\eta\|\mathcal{X}\|_1^2 \sum_{t=1}^{T-1} \|\boldsymbol{u}^{(t+1)} - \boldsymbol{u}^{(t)}\|_\infty^2 - \frac{1}{512\eta\|\mathcal{X}\|_1^2} \sum_{t=1}^{T-1} \|\boldsymbol{x}^{(t+1)} - \boldsymbol{x}^{(t)}\|_1^2.$$

*Proof.* At first, assume that $\|\boldsymbol{u}^{(t)}\|_\infty \leq 1/\|\mathcal{X}\|_1$. By definition of the induced dual local norm in (3),

$$\|\tilde{\boldsymbol{u}}^{(t)} - \tilde{\boldsymbol{u}}^{(t-1)}\|_{*,t}^2 \leq (\langle\boldsymbol{x}^{(t)}, \boldsymbol{u}^{(t)}\rangle - \langle\boldsymbol{x}^{(t-1)}, \boldsymbol{u}^{(t-1)}\rangle)^2 (\lambda^{(t)})^2 + \sum_{r=1}^{d} (\boldsymbol{y}[r])^2 (\boldsymbol{u}^{(t)}[r] - \boldsymbol{u}^{(t-1)}[r])^2$$

$$\leq (\langle\boldsymbol{x}^{(t)}, \boldsymbol{u}^{(t)}\rangle - \langle\boldsymbol{x}^{(t-1)}, \boldsymbol{u}^{(t-1)}\rangle)^2 + \sum_{r=1}^{d} (\boldsymbol{x}[r])^2 (\boldsymbol{u}^{(t)}[r] - \boldsymbol{u}^{(t-1)}[r])^2$$

$$\leq \left(\langle\boldsymbol{x}^{(t)}, \boldsymbol{u}^{(t)}\rangle - \langle\boldsymbol{x}^{(t-1)}, \boldsymbol{u}^{(t-1)}\rangle\right)^2 + \|\mathcal{X}\|_1^2 \|\boldsymbol{u}^{(t)} - \boldsymbol{u}^{(t-1)}\|_\infty^2, \tag{19}$$

for any $t \geq 2$. Further, by Young's inequality,

$$\left(\langle\boldsymbol{x}^{(t)}, \boldsymbol{u}^{(t)}\rangle - \langle\boldsymbol{x}^{(t-1)}, \boldsymbol{u}^{(t-1)}\rangle\right)^2 \leq 2\left(\langle\boldsymbol{x}^{(t)}, \boldsymbol{u}^{(t)} - \boldsymbol{u}^{(t-1)}\rangle\right)^2 + 2\left(\langle\boldsymbol{x}^{(t)} - \boldsymbol{x}^{(t-1)}, \boldsymbol{u}^{(t-1)}\rangle\right)^2$$

$$\leq 2\|\mathcal{X}\|_1^2 \|\boldsymbol{u}^{(t)} - \boldsymbol{u}^{(t-1)}\|_\infty^2 + \frac{2}{\|\mathcal{X}\|_1^2} \|\boldsymbol{x}^{(t)} - \boldsymbol{x}^{(t-1)}\|_1^2.$$

Combining with (19),

$$\|\tilde{\boldsymbol{u}}^{(t)} - \tilde{\boldsymbol{u}}^{(t-1)}\|_{*,t}^2 \leq 3\|\mathcal{X}\|_1^2 \|\boldsymbol{u}^{(t)} - \boldsymbol{u}^{(t-1)}\|_\infty^2 + \frac{2}{\|\mathcal{X}\|_1^2} \|\boldsymbol{x}^{(t)} - \boldsymbol{x}^{(t-1)}\|_1^2,$$

for $t \geq 2$, since $\|\boldsymbol{u}\|_\infty \leq \frac{1}{\|\mathcal{X}\|_1}$ (by assumption). Further, $\|\tilde{\boldsymbol{u}}^{(1)} - \tilde{\boldsymbol{u}}^{(0)}\|_{*,t}^2 = \|\tilde{\boldsymbol{u}}^{(1)}\|_{*,t}^2 \leq 2$. Combining with Proposition 2 and Lemma 1, we get that $\tilde{\mathrm{Reg}}^T$ is upper bounded by

$$6 + \frac{(d+1)\log T}{\eta} + 16\eta\|\mathcal{X}\|_1^2 \sum_{t=1}^{T-1} \|\boldsymbol{u}^{(t+1)} - \boldsymbol{u}^{(t)}\|_\infty^2 + \frac{1}{\|\mathcal{X}\|_1^2}\left(10\eta - \frac{1}{432\eta}\right)\sum_{t=1}^{T-1}\|\boldsymbol{x}^{(t+1)} - \boldsymbol{x}^{(t)}\|_1^2$$

$$\leq 6 + \frac{(d+1)\log T}{\eta} + 16\eta\|\mathcal{X}\|_1^2 \sum_{t=1}^{T-1}\|\boldsymbol{u}^{(t+1)} - \boldsymbol{u}^{(t)}\|_\infty^2 - \frac{1}{512\eta\|\mathcal{X}\|_1^2}\sum_{t=1}^{T-1}\|\boldsymbol{x}^{(t+1)} - \boldsymbol{x}^{(t)}\|_1^2.$$

Finally, we relax the assumption that $\|\boldsymbol{u}^{(t)}\|_\infty \leq 1/\|\mathcal{X}\|_1$. In that case, one can reduce to the above analysis by first rescaling all utilities by the factor $1/(B\|\mathcal{X}\|_1)$—which in turn is equivalent to rescaling the learning rate $\eta$ by $1/(B\|\mathcal{X}\|_1)$. We then need to correct for the fact that the norm of the difference of utilities gets rescaled by a factor $1/(B\|\mathcal{X}\|_1)^2$, and that the regret $\tilde{\mathrm{Reg}}^T$ with respect to the original utilities is a factor $B\|\mathcal{X}\|_1$ larger than the regret measured on the rescaled utilities. Taking these considerations into account leads to the statement. $\square$

### A.4   Main Result: Proof of Theorem 4

Finally, we are ready to establish Theorem 4. To this end, the main ingredient is the bound on the second-order path lengths predicted by Theorem 3, which is recalled below.

**Theorem 3.** *Suppose that Assumption 1 holds for some parameters $B, L > 0$. If all players follow* `LRL-OFTRL` *with learning rate $\eta \leq \min\left\{\frac{1}{256B\|\mathcal{X}\|_1}, \frac{1}{128nL\|\mathcal{X}\|_1^2}\right\}$, where $\|\mathcal{X}\|_1 := \max_{i\in[\![n]\!]}\|\mathcal{X}_i\|_1$, then*

$$\sum_{i=1}^n \sum_{t=1}^{T-1} \|\boldsymbol{x}_i^{(t+1)} - \boldsymbol{x}_i^{(t)}\|_1^2 \leq 6144n\eta B\|\mathcal{X}\|_1^3 + 1024n(d+1)\|\mathcal{X}\|_1^2 \log T. \tag{4}$$

*Proof.* By Assumption 1, it follows that for any player $i \in [\![n]\!]$,

$$\left(\|\boldsymbol{u}_i^{(t+1)} - \boldsymbol{u}_i^{(t)}\|_\infty\right)^2 \leq \left(L\sum_{j=1}^n \|\boldsymbol{x}_j^{(t+1)} - \boldsymbol{x}_j^{(t)}\|_1\right)^2 \leq L^2 n \sum_{j=1}^n \|\boldsymbol{x}_j^{(t+1)} - \boldsymbol{x}_j^{(t)}\|_1^2,$$

by Jensen's inequality. Hence, by Corollary 1 the regret $\mathrm{Reg}_i^T$ of each player $i \in [\![n]\!]$ can be upper bounded by

$$6B\|\mathcal{X}\|_1 + \frac{(d+1)\log T}{\eta} + 16\eta\|\mathcal{X}\|_1^2 L^2 n \sum_{j=1}^n\sum_{t=1}^{T-1}\|\boldsymbol{x}_j^{(t+1)} - \boldsymbol{x}_j^{(t)}\|_1^2 - \frac{1}{512\eta\|\mathcal{X}\|_1^2}\sum_{t=1}^{T-1}\|\boldsymbol{x}_i^{(t+1)} - \boldsymbol{x}_i^{(t)}\|_1^2,$$

Summing over all players $i \in [\![n]\!]$, we have that

$$\sum_{i=1}^n \tilde{\mathrm{Reg}}_i^T \leq 6nB\|\mathcal{X}\|_1 + n\frac{(d+1)\log T}{\eta} + \sum_{i=1}^n\left(16\eta\|\mathcal{X}\|_1^2 L^2 n^2 - \frac{1}{512\eta\|\mathcal{X}\|_1^2}\right)\sum_{t=1}^{T-1}\|\boldsymbol{x}_i^{(t+1)} - \boldsymbol{x}_i^{(t)}\|_1^2$$

$$\leq 6nB\|\mathcal{X}\|_1 + n\frac{(d+1)\log T}{\eta} - \frac{1}{1024\eta\|\mathcal{X}\|_1^2}\sum_{i=1}^n\sum_{t=1}^{T-1}\|\boldsymbol{x}_i^{(t+1)} - \boldsymbol{x}_i^{(t)}\|_1^2,$$

since $\eta \leq \frac{1}{256nL\|\mathcal{X}\|_1^2}$. Finally, the theorem follows since $\sum_{i=1}^n \tilde{\mathrm{Reg}}_i^T \geq 0$, which in turn follows directly from Theorem 2. $\square$

**Theorem 4** (Detailed Version of Theorem 1). *Suppose that Assumption 1 holds for some parameters $B, L > 0$. If all players follow* `LRL-OFTRL` *with learning rate $\eta = \min\left\{\frac{1}{256B\|\mathcal{X}\|_1}, \frac{1}{128nL\|\mathcal{X}\|_1^2}\right\}$, then for any $T \in \mathbb{N}$ the regret $\mathrm{Reg}_i^T$ of each player $i \in [\![n]\!]$ can be bounded as*

$$\mathrm{Reg}_i^T \leq 12B\|\mathcal{X}\|_1 + 256(d+1)\max\left\{nL\|\mathcal{X}\|_1^2, 2B\|\mathcal{X}\|_1\right\}\log T. \tag{5}$$

*Furthermore, the algorithm can be adaptive so that if player $i$ is instead facing adversarial utilities, then $\mathrm{Reg}_i^T = O(\sqrt{T})$.*

*Proof.* First of all, by Assumption 1 we have that for any player $i \in [\![n]\!]$,

$$\|\boldsymbol{u}_i^{(t+1)} - \boldsymbol{u}_i^{(t)}\|_\infty^2 \leq \left(L \sum_{j=1}^n \|\boldsymbol{x}_j^{(t+1)} - \boldsymbol{x}_j^{(t)}\|_1\right)^2 \leq L^2 n \sum_{j=1}^n \|\boldsymbol{x}_j^{(t+1)} - \boldsymbol{x}_j^{(t)}\|_1^2.$$

Hence, summing over all $t$,

$$\sum_{t=1}^{T-1} \|\boldsymbol{u}_i^{(t+1)} - \boldsymbol{u}_i^{(t)}\|_\infty^2 \leq L^2 n \sum_{t=1}^{T-1} \sum_{j=1}^n \|\boldsymbol{x}_j^{(t+1)} - \boldsymbol{x}_j^{(t)}\|_1^2$$

$$\leq 6144 n^2 L^2 \eta B \|\mathcal{X}\|_1^3 + 1024 n^2 L^2 (d+1) \|\mathcal{X}\|_1^2 \log T,$$

where the last bound uses Theorem 3. As a result, from Corollary 1, if $\eta = \frac{1}{128 n L \|\mathcal{X}\|_1^2}$,

$$\tilde{\mathrm{Reg}}_i^T \leq 6B\|\mathcal{X}\|_1 + \frac{(d+1)\log T}{\eta} + 16\eta\|\mathcal{X}\|_1^2 \sum_{t=1}^{T-1} \|\boldsymbol{u}_i^{(t+1)} - \boldsymbol{u}_i^{(t)}\|_\infty^2$$

$$\leq 12B\|\mathcal{X}\|_1 + 256(d+1)nL\|\mathcal{X}\|_1^2 \log T.$$

Thus, the bound on $\mathrm{Reg}_i^T$ follows directly since $\mathrm{Reg}_i^T \leq \tilde{\mathrm{Reg}}_i^T$ by Theorem 2. The case where $\eta = \frac{1}{256 B \|\mathcal{X}\|_1}$ is analogous.

Next, let us focus on the adversarial bound. Each player can simply check whether there exists a time $t \in [\![T]\!]$ such that

$$\sum_{\tau=1}^{t-1} \|\boldsymbol{u}_i^{(\tau+1)} - \boldsymbol{u}_i^{(\tau)}\|_\infty^2 > 6144 n^2 L^2 \eta B \|\mathcal{X}\|_1^3 + 1024 n^2 L^2 (d+1) \|\mathcal{X}\|_1^2 \log t. \qquad (20)$$

In particular, we know from Theorem 3 that when all players follow the prescribed protocol (20) will never by satisfied. On the other hand, if there exists time $t$ so that (20) holds, then it suffices to switch to any no-regret learning algorithm tuned to face adversarial utilities. $\qquad\square$

### A.5 Extending the Analysis under Approximate Iterates

In this subsection we describe how to extend our analysis, and in particular Theorem 4, when the `OFTRL` step of Algorithm 1 at time $t$ is only computed with tolerance $\epsilon^{(t)}$, in the sense of (6). We start by extending Proposition 2 below.

**Proposition 5** (Extension of Proposition 2). *Let $\tilde{\mathrm{Reg}}^T$ be the regret cumulated up to time $T$ by the internal `OFTRL` algorithm producing approximate iterates $(\lambda^{(t)}, \boldsymbol{y}^{(t)}) \in \tilde{\mathcal{X}}$, for any $t \in [\![T]\!]$. Then, for any $T \in \mathbb{N}$ and learning rate $\eta \leq \frac{1}{50}$,*

$$\tilde{\mathrm{Reg}}^T \leq 4 + \frac{(d+1)\log T}{\eta} + 5\eta \sum_{t=1}^T \left\|\tilde{\boldsymbol{u}}^{(t)} - \tilde{\boldsymbol{u}}^{(t-1)}\right\|_{*,t}^2 - \frac{1}{27\eta} \sum_{t=1}^{T-1} \left\|\begin{pmatrix}\lambda_\star^{(t+1)}\\\boldsymbol{y}_\star^{(t+1)}\end{pmatrix} - \begin{pmatrix}\lambda_\star^{(t)}\\\boldsymbol{y}_\star^{(t)}\end{pmatrix}\right\|_{(\lambda_\star^{(t)},\boldsymbol{y}_\star^{(t)})}$$

$$+ 2\sum_{t=1}^T \left\|\begin{pmatrix}\lambda^{(t)}\\\boldsymbol{y}^{(t)}\end{pmatrix} - \begin{pmatrix}\lambda_\star^{(t)}\\\boldsymbol{y}_\star^{(t)}\end{pmatrix}\right\|_{(\lambda_\star^{(t)},\boldsymbol{y}_\star^{(t)})},$$

*where*

$$\begin{pmatrix}\lambda_\star^{(t)}\\\boldsymbol{y}_\star^{(t)}\end{pmatrix} := \arg\max_{(\lambda,\boldsymbol{y})\in\tilde{\mathcal{X}}}\left\{\eta\left\langle \tilde{\boldsymbol{U}}^{(t)} + \tilde{\boldsymbol{u}}^{(t-1)}, \begin{pmatrix}\lambda\\\boldsymbol{y}\end{pmatrix}\right\rangle + \log\lambda + \sum_{r=1}^d \log\boldsymbol{y}[r]\right\}.$$

*Proof.* Fix any $(\lambda^*, \boldsymbol{y}^*) \in \tilde{\mathcal{X}}$. Then,

$$\sum_{t=1}^T \left\langle \tilde{\boldsymbol{u}}^{(t)}, \begin{pmatrix}\lambda^*\\\boldsymbol{y}^*\end{pmatrix} - \begin{pmatrix}\lambda^{(t)}\\\boldsymbol{y}^{(t)}\end{pmatrix}\right\rangle = \sum_{t=1}^T \left\langle \tilde{\boldsymbol{u}}^{(t)}, \begin{pmatrix}\lambda^*\\\boldsymbol{y}^*\end{pmatrix} - \begin{pmatrix}\lambda_\star^{(t)}\\\boldsymbol{y}_\star^{(t)}\end{pmatrix}\right\rangle + \sum_{t=1}^T \left\langle \tilde{\boldsymbol{u}}^{(t)}, \begin{pmatrix}\lambda_\star^{(t)}\\\boldsymbol{y}_\star^{(t)}\end{pmatrix} - \begin{pmatrix}\lambda^{(t)}\\\boldsymbol{y}^{(t)}\end{pmatrix}\right\rangle$$

$$\leq \sum_{t=1}^T \left\langle \tilde{\boldsymbol{u}}^{(t)}, \begin{pmatrix}\lambda^*\\\boldsymbol{y}^*\end{pmatrix} - \begin{pmatrix}\lambda_\star^{(t)}\\\boldsymbol{y}_\star^{(t)}\end{pmatrix}\right\rangle + 2\sum_{t=1}^T \left\|\begin{pmatrix}\lambda^{(t)}\\\boldsymbol{y}^{(t)}\end{pmatrix} - \begin{pmatrix}\lambda_\star^{(t)}\\\boldsymbol{y}_\star^{(t)}\end{pmatrix}\right\|_{(\lambda_\star^{(t)},\boldsymbol{y}_\star^{(t)})},$$

where the last inequality uses Hölder's inequality along with the fact that $\|\tilde{\boldsymbol{u}}^{(t)}\|_{*,(\lambda^{(t)},\boldsymbol{y}^{(t)})} \le 2$, which in turn follows since $\|\boldsymbol{u}^{(t)}\|_\infty \|\mathcal{X}\|_1 \le 1$ (by assumption). Finally, the proof follows as an immediate consequence of Proposition 2. $\qquad\square$

We next proceed with the extension of Lemma 1.

**Lemma 7** (Extension of Lemma 1). *Suppose that $\epsilon^{(t)} \le \frac{1}{8}$, for any $t \in \llbracket T \rrbracket$. Then, for any time $t \in \llbracket T-1 \rrbracket$ and learning rate $\eta \le \frac{1}{256}$,*

$$\|\boldsymbol{x}^{(t+1)} - \boldsymbol{x}^{(t)}\|_1 \le 8\|\mathcal{X}\|_1 \left\| \begin{pmatrix} \lambda_\star^{(t+1)} \\ \boldsymbol{y}_\star^{(t+1)} \end{pmatrix} - \begin{pmatrix} \lambda_\star^{(t)} \\ \boldsymbol{y}_\star^{(t)} \end{pmatrix} \right\|_{(\lambda_\star^{(t)},\boldsymbol{y}_\star^{(t)})} + 16\|\mathcal{X}\|_1 \epsilon^{(t+1)} + 8\|\mathcal{X}\|_1 \epsilon^{(t)},$$

*where $\boldsymbol{x}^{(t)} := \boldsymbol{y}^{(t)}/\lambda^{(t)}$.*

*Proof.* First, by the triangle inequality,

$$\left\| \begin{pmatrix} \lambda_\star^{(t+1)} \\ \boldsymbol{y}_\star^{(t+1)} \end{pmatrix} - \begin{pmatrix} \lambda_\star^{(t)} \\ \boldsymbol{y}_\star^{(t)} \end{pmatrix} \right\|_{(\lambda_\star^{(t)},\boldsymbol{y}_\star^{(t)})} \ge \left\| \begin{pmatrix} \lambda^{(t+1)} \\ \boldsymbol{y}^{(t+1)} \end{pmatrix} - \begin{pmatrix} \lambda^{(t)} \\ \boldsymbol{y}^{(t)} \end{pmatrix} \right\|_{(\lambda_\star^{(t)},\boldsymbol{y}_\star^{(t)})}$$
$$- \left\| \begin{pmatrix} \lambda_\star^{(t+1)} \\ \boldsymbol{y}_\star^{(t+1)} \end{pmatrix} - \begin{pmatrix} \lambda^{(t+1)} \\ \boldsymbol{y}^{(t+1)} \end{pmatrix} \right\|_{(\lambda_\star^{(t)},\boldsymbol{y}_\star^{(t)})} - \left\| \begin{pmatrix} \lambda_\star^{(t)} \\ \boldsymbol{y}_\star^{(t)} \end{pmatrix} - \begin{pmatrix} \lambda^{(t)} \\ \boldsymbol{y}^{(t)} \end{pmatrix} \right\|_{(\lambda_\star^{(t)},\boldsymbol{y}_\star^{(t)})}.$$

Now given that $\eta \le \frac{1}{50}$, it follows from Proposition 3 that

$$\left\| \begin{pmatrix} \lambda_\star^{(t+1)} \\ \boldsymbol{y}_\star^{(t+1)} \end{pmatrix} - \begin{pmatrix} \lambda_\star^{(t)} \\ \boldsymbol{y}_\star^{(t)} \end{pmatrix} \right\|_{(\lambda_\star^{(t)},\boldsymbol{y}_\star^{(t)})} \le \frac{1}{2},$$

which in turn—combined with Lemma 4—implies that

$$\left\| \begin{pmatrix} \lambda_\star^{(t+1)} \\ \boldsymbol{y}_\star^{(t+1)} \end{pmatrix} - \begin{pmatrix} \lambda^{(t+1)} \\ \boldsymbol{y}^{(t+1)} \end{pmatrix} \right\|_{(\lambda_\star^{(t)},\boldsymbol{y}_\star^{(t)})} \le 2 \left\| \begin{pmatrix} \lambda_\star^{(t+1)} \\ \boldsymbol{y}_\star^{(t+1)} \end{pmatrix} - \begin{pmatrix} \lambda^{(t+1)} \\ \boldsymbol{y}^{(t+1)} \end{pmatrix} \right\|_{(\lambda_\star^{(t+1)},\boldsymbol{y}_\star^{(t+1)})}.$$

Similarly, since $\epsilon^{(t)} \le \frac{1}{8}$, it follows that

$$\left\| \begin{pmatrix} \lambda^{(t+1)} \\ \boldsymbol{y}^{(t+1)} \end{pmatrix} - \begin{pmatrix} \lambda^{(t)} \\ \boldsymbol{y}^{(t)} \end{pmatrix} \right\|_{(\lambda_\star^{(t)},\boldsymbol{y}_\star^{(t)})} \ge \frac{1}{2} \left\| \begin{pmatrix} \lambda^{(t+1)} \\ \boldsymbol{y}^{(t+1)} \end{pmatrix} - \begin{pmatrix} \lambda^{(t)} \\ \boldsymbol{y}^{(t)} \end{pmatrix} \right\|_{(\lambda^{(t)},\boldsymbol{y}^{(t)})}.$$

As a result,

$$\left\| \begin{pmatrix} \lambda_\star^{(t+1)} \\ \boldsymbol{y}_\star^{(t+1)} \end{pmatrix} - \begin{pmatrix} \lambda_\star^{(t)} \\ \boldsymbol{y}_\star^{(t)} \end{pmatrix} \right\|_{(\lambda_\star^{(t)},\boldsymbol{y}_\star^{(t)})} \ge \frac{1}{2} \left\| \begin{pmatrix} \lambda^{(t+1)} \\ \boldsymbol{y}^{(t+1)} \end{pmatrix} - \begin{pmatrix} \lambda^{(t)} \\ \boldsymbol{y}^{(t)} \end{pmatrix} \right\|_{(\lambda^{(t)},\boldsymbol{y}^{(t)})} - 2\epsilon^{(t+1)} - \epsilon^{(t)}. \quad (21)$$

Next, we will prove that

$$\max\left\{ \left| \frac{\lambda^{(t+1)}}{\lambda^{(t)}} - 1 \right|, \max_{r \in \llbracket d \rrbracket} \left| \frac{\boldsymbol{y}^{(t+1)}[r]}{\boldsymbol{y}^{(t)}[r]} - 1 \right| \right\} \le \frac{1}{2}. \quad (22)$$

Indeed, since $\epsilon^{(t)}, \epsilon^{(t+1)} \le \frac{1}{8}$, it holds that

$$\left| 1 - \frac{\lambda^{(t)}}{\lambda_\star^{(t)}} \right| \le \frac{1}{8} \implies \frac{7}{8}\lambda_\star^{(t)} \le \lambda^{(t)} \le \frac{9}{8}\lambda_\star^{(t)},$$

and

$$\left| 1 - \frac{\lambda^{(t+1)}}{\lambda_\star^{(t+1)}} \right| \le \frac{1}{8} \implies \frac{7}{8}\lambda_\star^{(t+1)} \le \lambda^{(t+1)} \le \frac{9}{8}\lambda_\star^{(t+1)}.$$

Furthermore, for $\eta \le \frac{1}{256}$,

$$\left| 1 - \frac{\lambda_\star^{(t+1)}}{\lambda_\star^{(t)}} \right| \le \frac{1}{10} \implies \frac{9}{10}\lambda_\star^{(t)} \le \lambda_\star^{(t+1)} \le \frac{11}{10}\lambda_\star^{(t)},$$

by Proposition 3 and Lemma 4. Thus,

$$\frac{2}{3}\lambda^{(t+1)} \le \frac{7}{8}\frac{10}{11}\frac{8}{9}\lambda^{(t+1)} \le \lambda^{(t)} \le \frac{9}{8}\frac{10}{9}\frac{8}{7}\lambda^{(t+1)} \le \frac{3}{2}\lambda^{(t+1)},$$

in turn implying that

$$\left| 1 - \frac{\lambda^{(t+1)}}{\lambda^{(t)}} \right| \le \frac{1}{2}.$$

Similarly, we conclude that for any $r \in [\![d]\!]$,

$$\left| 1 - \frac{\boldsymbol{y}^{(t+1)}[r]}{\boldsymbol{y}^{(t)}[r]} \right| \le \frac{1}{2},$$

confirming (22). Hence, following the proof of Lemma 6, we derive that

$$\left\| \begin{pmatrix} \lambda^{(t+1)} \\ \boldsymbol{y}^{(t+1)} \end{pmatrix} - \begin{pmatrix} \lambda^{(t)} \\ \boldsymbol{y}^{(t)} \end{pmatrix} \right\|_{(\lambda^{(t)}, \boldsymbol{y}^{(t)})} \ge \frac{1}{4\|\mathcal{X}\|_1} \left\| \frac{\boldsymbol{y}^{(t+1)}}{\lambda^{(t+1)}} - \frac{\boldsymbol{y}^{(t)}}{\lambda^{(t)}} \right\|_1 = \frac{1}{4\|\mathcal{X}\|_1} \|\boldsymbol{x}^{(t+1)} - \boldsymbol{x}^{(t)}\|_1.$$

Combining this bound with (21) concludes the proof. $\qquad\square$

We also state the following immediate implication of Lemma 7.

**Corollary 3.** *Suppose that $\epsilon^{(t)} \le \frac{1}{8}$, for any $t \in [\![T]\!]$. Then, for any $t \in [\![T-1]\!]$ and learning rate $\eta \le \frac{1}{256}$,*

$$\|\boldsymbol{x}^{(t+1)} - \boldsymbol{x}^{(t)}\|_1^2 \le 192\|\mathcal{X}\|_1^2 \left\| \begin{pmatrix} \lambda_\star^{(t+1)} \\ \boldsymbol{y}_\star^{(t+1)} \end{pmatrix} - \begin{pmatrix} \lambda_\star^{(t)} \\ \boldsymbol{y}_\star^{(t)} \end{pmatrix} \right\|_{(\lambda_\star^{(t)}, \boldsymbol{y}_\star^{(t)})}^2 + 768\|\mathcal{X}\|_1^2 (\epsilon^{(t+1)})^2 + 192\|\mathcal{X}\|_1^2 (\epsilon^{(t)})^2,$$

*where $\boldsymbol{x}^{(t)} := \boldsymbol{y}^{(t)} / \lambda^{(t)}$.*

As a result, combining this bound with Proposition 5 extends Corollary 1 with an error term proportional to $\sum_{t=1}^{T} \epsilon^{(t)}$. Finally, the rest of the extension is identical to our proof of Theorem 4.

# B   Implementation via Proximal Oracles

In this section we provide the omitted proofs from Section 3.5 regarding the implementation of LRL-OFTRL using proximal oracles (recall Equation (7)).

## B.1   The Proximal Newton Method

In this subsection we describe the proximal Newton algorithm of Tran-Dinh et al. [2015], leading to Theorem 5 we presented in Section 3.5. More precisely, Tran-Dinh et al. [2015] studied the following composite minimization problem:

$$\min_{\tilde{\boldsymbol{x}} \in \mathbb{R}^{d+1}} \left\{ F(\tilde{\boldsymbol{x}}) := f(\tilde{\boldsymbol{x}}) + g(\tilde{\boldsymbol{x}}) \right\}, \tag{23}$$

where $f$ is a (standard) self-concordant and convex function, and $g : \mathbb{R}^{d+1} \to \mathbb{R} \cup \{+\infty\}$ is a proper, closed and convex function. In our setting, we will let $g$ be defined as

$$g(\tilde{\boldsymbol{x}}) := \begin{cases} 0 & \text{if } \tilde{\boldsymbol{x}} \in \tilde{\mathcal{X}}, \\ +\infty & \text{otherwise.} \end{cases}$$

Further, for a given time $t \in \mathbb{N}$, we let

$$f : \tilde{\boldsymbol{x}} \mapsto -\eta \left\langle \tilde{\boldsymbol{U}}^{(t)} + \tilde{\boldsymbol{u}}^{(t-1)}, \tilde{\boldsymbol{x}} \right\rangle - \sum_{r=1}^{d+1} \log \tilde{\boldsymbol{x}}[r].$$

Before we describe the proximal Newton method, let us define $\tilde{\boldsymbol{s}}_k$ as follows.

$$\tilde{\boldsymbol{s}}_k := \arg\min_{\tilde{\boldsymbol{x}} \in \tilde{\mathcal{X}}} \left\{ f(\tilde{\boldsymbol{x}}_k) + (\nabla f(\tilde{\boldsymbol{x}}_k))^\top (\tilde{\boldsymbol{x}} - \tilde{\boldsymbol{x}}_k) + \frac{1}{2}(\tilde{\boldsymbol{x}} - \tilde{\boldsymbol{x}}_k)^\top \nabla^2 f(\tilde{\boldsymbol{x}}_k)(\tilde{\boldsymbol{x}} - \tilde{\boldsymbol{x}}_k) \right\}, \quad (24)$$

for some $\tilde{\boldsymbol{x}}_k \in \mathbb{R}^{d+1}_{>0}$. We point out that the optimization problem (24) can be trivially solved when we have access to a (local) proximal oracle—given in Equation (7).

In this context, the proximal Newton method of Tran-Dinh et al. [2015] is given in Algorithm 2. Their algorithm proceeds in two phases. In the first phase we perform *damped steps* of proximal Newton until we reach the region of quadratic convergence. Afterwards, we perform *full steps* of proximal Newton until the desired precision $\epsilon > 0$ has been reached. Below we summarize the main guarantee regarding Algorithm 2, namely [Tran-Dinh et al., 2015, Theorem 9].

**Theorem 7** ( [Tran-Dinh et al., 2015]). *Algorithm 2 returns $\tilde{\boldsymbol{x}}_K \in \mathbb{R}^{d+1}_{>0}$ such that $\|\tilde{\boldsymbol{x}}_K - \tilde{\boldsymbol{x}}^*\|_{\tilde{\boldsymbol{x}}^*} \leq 2\epsilon$ after at most*

$$K = \left\lfloor \frac{f(\tilde{\boldsymbol{x}}_0) - f(\tilde{\boldsymbol{x}}^*)}{0.017} \right\rfloor + \left\lfloor 1.5 \ln \ln \left( \frac{0.28}{\epsilon} \right) \right\rfloor + 2$$

*iterations, for any $\epsilon > 0$, where $\tilde{\boldsymbol{x}}^* = \arg\min_{\tilde{\boldsymbol{x}}} F(\tilde{\boldsymbol{x}})$, for the composite function $F$ defined in (23).*

To establish Theorem 5 from this guarantee, it suffices to initialize Algorithm 2 at every iteration $t \geq 2$ with $\tilde{\boldsymbol{x}}_0 := \tilde{\boldsymbol{x}}^{(t-1)} = (\lambda^{(t-1)}, \boldsymbol{y}^{(t-1)})$. Then, as long as $\epsilon^{(t-1)}$ is sufficiently small, the number of iterations predicted by Theorem 7 will be bounded by $O(\log \log(1/\epsilon))$, in turn establishing Theorem 5.

---

**Algorithm 2:** Proximal Newton [Tran-Dinh et al., 2015]

---

**Data:** Initial point $\tilde{\boldsymbol{x}}_0$
Precision $\epsilon > 0$
Constant $\sigma := 0.2$

1  **for** $k = 1, \ldots, K$ **do**
2  |  Obtain the proximal Newton direction $\tilde{\boldsymbol{d}}_k \leftarrow \tilde{\boldsymbol{s}}_k - \tilde{\boldsymbol{x}}_k$, where $\tilde{\boldsymbol{s}}_k$ is defined in (24)
3  |  Set $\lambda_k \leftarrow \|\tilde{\boldsymbol{d}}_k\|_{\tilde{\boldsymbol{x}}_k}$
4  |  **if** $\lambda_k > 0.2$ **then**
5  |  |  $\tilde{\boldsymbol{x}}_{k+1} \leftarrow \tilde{\boldsymbol{x}}_k + \alpha_k \tilde{\boldsymbol{d}}_k$, where $\alpha_k := (1 + \lambda_k)^{-1}$          [▷ Damped Step]
6  |  **else if** $\lambda_k > \epsilon$ **then**
7  |  |  $\tilde{\boldsymbol{x}}_{k+1} \leftarrow \tilde{\boldsymbol{x}}_k + \tilde{\boldsymbol{d}}_k$          [▷ Full Step]
8  |  **else**
9  |  |  **return** $\tilde{\boldsymbol{x}}_k$

---

### B.2 Proximal Oracle for Normal-Form and Extensive-Form Games

In order to show that the proximal oracle of Section 3.5 can be implemented efficiently for probability simplexes (*i.e.*, the strategy sets of normal-form games) and sequence-form strategy spaces (*i.e.*, the strategy sets of extensive-form games), we will prove a slightly stronger result concerning *treeplex* sets, of which sequence-form strategy spaces are instances.

**Definition 1.** *A set $Q \subseteq [0, +\infty)^d$, $d \geq 1$, is* treeplex *if it is:*

1. *a simplex $Q = \Delta^d$;*

2. *a Cartesian product of treeplex sets $Q_1 \times \cdots \times Q_K$; or*

3. *(Branching operation) a set of the form*

$$\triangle(Q_1, \ldots, Q_K) := \{(\boldsymbol{x}, \boldsymbol{x}[1]\boldsymbol{q}_1, \ldots, \boldsymbol{x}[K]\boldsymbol{q}_K) : \boldsymbol{x} \in \Delta^K, \boldsymbol{q}_k \in Q_k \ \forall k \in [\![K]\!]\},$$

*where $Q_1, \ldots, Q_K$ are treeplex.*

We will show that any treeplex $Q$ is such that $[0, 1]Q$ admits an efficient (positive-definite) quadratic optimization oracle. This is sufficient, since it is well-known that every sequence-form strategy space $\mathcal{X}$ is treeplex (e.g., Hoda et al. [2010]) and therefore, by definition, so is the set $\{(1, \boldsymbol{x}) : \boldsymbol{x} \in \mathcal{X}\}$.

Introduce the *value function*

$$V_Q(t; \boldsymbol{g}, \boldsymbol{w}) := \min_{\boldsymbol{x} \in tQ} \left\{ -\boldsymbol{g}^\top \boldsymbol{x} + \frac{1}{2} \sum_{r=1}^d \left( \frac{\boldsymbol{x}[r]}{\boldsymbol{w}[r]} \right)^2 \right\} \qquad (t \geq 0, \boldsymbol{w} > \boldsymbol{0}) \qquad (25)$$

(note the rescaling by $t$ in the domain of the minimization). We will be interested in the derivative of $V_Q(t; \boldsymbol{g}, \boldsymbol{w})$, which we will denote as[6]

$$\lambda_Q(t; \boldsymbol{g}, \boldsymbol{w}) := \frac{d}{dt} V_Q(t; \boldsymbol{g}, \boldsymbol{w}).$$

**Preliminaries on Strictly Monotonic Piecewise-Linear (SMPL) Functions**

**Definition 2** (SMPL function and standard representation). *Given an interval $I \subseteq \mathbb{R}$ and a function $f : I \to \mathbb{R}$, we say that $f$ is SMPL if it is strictly monotonically increasing and piecewise-linear on $I$.*

**Definition 3** (Quasi-SMPL function). *A quasi-SMPL function is a function $f : \mathbb{R} \to [0, +\infty)$ of the form $f(x) = [g(x)]^+$ where $g(x) : \mathbb{R} \to \mathbb{R}$ is SMPL and $[\,\cdot\,]^+ := \max\{0, \cdot\}$.*

**Definition 4.** *Given a SMPL or quasi-SMPL function $f$, a* standard representation *for it is an expression of the form*

$$f(x) = \zeta + \alpha_0 x + \sum_{s=1}^S \alpha_s [x - \beta_s]^+,$$

*valid for all $x$ in the domain of $f$, where $S \in \mathbb{N} \cup \{0\}$ and $\beta_1 < \cdots < \beta_S$. The size of the standard representation is defined as the natural number $S$.*

We now mention four basic results about SMPL and quasi-SMPL functions. The proofs are elementary and omitted.

**Lemma 8.** *Let $f : I \to \mathbb{R}$ be SMPL, and consider a standard representation of $f$ of size $S$. Then, for any $\zeta \in \mathbb{R}$ and $\alpha \geq 0$, a standard representation for the SMPL function $I \ni x \mapsto \zeta + \alpha f(x)$ can be computed in $O(S + 1)$ time.*

**Lemma 9.** *The sum $f_1 + \cdots + f_n$ of $n$ SMPL (resp., quasi-SMPL) functions $f_i : I \to \mathbb{R}$ is a SMPL (resp., quasi-SMPL) function $I \to \mathbb{R}$. Furthermore, if each $f_i$ admits a standard representation of size $S_i$, then a standard representation of size at most $S_1 + \cdots + S_n$ for their sum can be computed in $O((S_1 + \cdots + S_n + 1) \log n)$ time.*

**Lemma 10.** *Let $f : \mathbb{R} \to \mathbb{R}$ be SMPL, and consider a standard representation of $f$ of size $S$. Then, for any $\beta \in \mathbb{R}$, a standard representation of size at most $S$ for the quasi-SMPL function $I \ni x \mapsto [f(x) - \beta]^+$ can be computed in $O(S + 1)$ time.*

**Lemma 11.** *The inverse $f^{-1} : \text{range}(f) \to \mathbb{R}$ of a SMPL function $f : I \to \mathbb{R}$ is SMPL. Furthermore, if $f$ admits a standard representation of size $S$, then a standard representation for $f^{-1}$ of size at most $S$ can be computed in $O(S + 1)$ time.*

**Lemma 12.** *Let $f : \mathbb{R} \to [0, +\infty)$ be quasi-SMPL. The restricted inverse $f^{-1} : (0, +\infty) \to \mathbb{R}$ of $f$ is SMPL, where we restrict the domain to $(0, +\infty)$ because $f^{-1}(0)$ may be multivalued. Furthermore, if $f$ admits a standard representation of size $S$, then a standard representation of size at most $S$ for $f^{-1}$ can be computed in $O(S + 1)$ time.*

*Proof.* We have $f(x) = [g(x)]^+$ where $g$ is SMPL. It follows that the function $\bar{g} : I \to \mathbb{R}$ defined as $\bar{g}(x) = g(x)$ for the interval $I = \{x : g(x) > 0\}$ is SMPL as well. For any $x$ such that $f(x) > 0$ we have $x \in I$, and thus $f^{-1} = g^{-1}$, and it follows from Lemma 11 that $f^{-1}$ is SMPL. $\qquad \square$

_______________

[6]For $t = 0$ we define $\lambda_Q(t; \boldsymbol{g}, \boldsymbol{w})$ in the usual way as

$$\lambda_Q(0; \boldsymbol{g}, \boldsymbol{w}) = \lim_{t \to 0^+} \frac{V_Q(t; \boldsymbol{g}, \boldsymbol{w}) - V_Q(0; \boldsymbol{g}, \boldsymbol{w})}{t} = \lim_{t \to 0^+} \frac{V_Q(t; \boldsymbol{g}, \boldsymbol{w})}{t}.$$

**Lemma 13.** *Let $f : [0, +\infty) \to \mathbb{R}$ be a SMPL function, and consider the function $g$ that maps $y$ to the unique solution to the equation $x = [y - f(x)]^+$. Then, $g$ is quasi-SMPL and satisfies $g(y) = [(x+f)^{-1}(y)]^+$, where $(x+f)^{-1}$ denotes the inverse of the SMPL function $x \mapsto x + f([x]^+)$.*

*Proof.* For any $y \in \mathbb{R}$, the function $h_y : x \mapsto x - [y - f(x)]^+$ is clearly SMPL on $[0, +\infty)$. Furthermore, $h_y(0) \leq 0$ and $h_y(+\infty) = +\infty$, implying that $h_y(x) = 0$ has a unique solution. We now show that $g(y) = [(x + f)^{-1}(y)]^+$ is that solution, that is, it satisfies $g(y) = [y - f(g(y))]^+$ for all $y \in \mathbb{R}$. Fix any $y \in \mathbb{R}$ and let

$$\bar{g} := (x+f)^{-1}(y) \quad \Longleftrightarrow \quad \bar{g} + f([\bar{g}]^+) = y \quad \Longleftrightarrow \quad \bar{g} = y - f([\bar{g}]^+) \tag{26}$$

There are two cases:

- If $\bar{g} \geq 0$, then $g(y) = [\bar{g}]^+ = \bar{g}$, and so we have

$$g(y) = [\bar{g}]^+ = [y - f([\bar{g}]^+)]^+ = [y - f(g(y))]^+,$$

as we wanted to show.

- Otherwise, $\bar{g} < 0$ and $g(y) = 0$. From (26), the condition $\bar{g} < 0$ implies $y < f([\bar{g}]^+) = f(0)$. So, it is indeed the case that

$$0 = g(y) = [y - f(0)]^+ = [y - f(g(0))]^+,$$

as we wanted to show.

Finally, we note that the function $(x + f)^{-1} : \mathbb{R} \to \mathbb{R}$ is SMPL due to Lemma 11, implying that $g(y)$ is quasi-SMPL. $\square$

**Central result** The following result is central in our analysis.

**Lemma 14.** *For any treeplex $Q \subseteq \mathbb{R}^d$, gradient $\boldsymbol{g} \in \mathbb{R}^d$, and center $\boldsymbol{w} \in \mathbb{R}^d_{>0}$, the function $t \mapsto \lambda_Q(t; \boldsymbol{g}, \boldsymbol{w})$ is SMPL, and a standard representation of it of size $d$ can be computed in polynomial time in $d$.*

*Proof.* We will prove the result by structural induction on $Q$.

- First, we consider the case where $Q$ is a Cartesian product,

$$Q = Q_1 \times \cdots \times Q_K.$$

In that case, the value function decomposes as follows

$$V_Q(t; \boldsymbol{g}, \boldsymbol{w}) = \sum_{k=1}^K \min_{\boldsymbol{x}_k \in tQ_k} \left\{ -\boldsymbol{g}_k^\top \boldsymbol{x}_k + \frac{1}{2} \sum_{r=1}^{d_k} \left( \frac{\boldsymbol{x}_k[r]}{\boldsymbol{w}_k[r]} \right)^2 \right\} = \sum_{k=1}^K V_{Q_k}(t; \boldsymbol{g}_k, \boldsymbol{w}_k).$$

By linearity of derivatives, we have

$$\lambda_Q(t; \boldsymbol{g}, \boldsymbol{w}) = \sum_{k=1}^K \lambda_{Q_k}(t; \boldsymbol{g}_k, \boldsymbol{w}_k).$$

From Lemma 9, we conclude that $\lambda_Q(t; \boldsymbol{g}, \boldsymbol{w})$ is a SMPL function with domain $[0, +\infty)$ which admits a standard representation of size at most $d = d_1 + \cdots + d_K$ computable in time $O(d \log K)$ starting from the standard representation of each of the $\lambda_{Q_k}(t; \boldsymbol{g}_k, \boldsymbol{w}_k)$.

- Second, consider the case where $Q$ is a simplex or the result of a branching operation

$$\triangle(Q_1, \ldots, Q_K) = \{(\boldsymbol{x}, \boldsymbol{x}[1]\boldsymbol{q}_1, \ldots, \boldsymbol{x}[K]\boldsymbol{q}_K) : \boldsymbol{x} \in \Delta^K, \boldsymbol{q}_k \in Q_k \; \forall k \in [\![K]\!]\},$$

where $Q_k \in \mathbb{R}^{d_k}$. With a slighty abuse of notation, we will treat the two cases together, considering the $K$-simplex $\Delta^K$ as a branching operation over empty sets $Q_k = \emptyset$.

In this case, we can write

$$\boldsymbol{g} = (\boldsymbol{g}_\bullet[1], \ldots, \boldsymbol{g}_\bullet[K], \boldsymbol{g}_1 \in \mathbb{R}^{d_1}, \cdots, \boldsymbol{g}_K \in \mathbb{R}^{d_K}), \text{ and}$$

$$\boldsymbol{w} = (\boldsymbol{w}_\bullet[1], \ldots, \boldsymbol{w}_\bullet[K], \boldsymbol{w}_1 \in \mathbb{R}^{d_1}_{>0}, \cdots, \boldsymbol{w}_K \in \mathbb{R}^{d_K}_{>0}).$$

The value function then decomposes recursively as

$$
V_Q(t; \boldsymbol{g}, \boldsymbol{w}) = \min_{\boldsymbol{x}_\bullet \in t\Delta^K} \left\{ \left( -\sum_{r=1}^K \boldsymbol{g}_\bullet[r]\boldsymbol{x}_\bullet[r] + \frac{1}{2}\sum_{r=1}^K \left(\frac{\boldsymbol{x}_\bullet[r]}{\boldsymbol{w}_\bullet[r]}\right)^2 \right) \right.
$$
$$
\left. + \sum_{k=1}^K \min_{\boldsymbol{x}_k \in \boldsymbol{x}_\bullet[k]Q_k} \left\{ -\boldsymbol{g}_k^\top \boldsymbol{x}_k + \sum_{r=1}^{d_k} \left(\frac{\boldsymbol{x}_k[r]}{\boldsymbol{w}_k[r]}\right)^2 \right\} \right\}
$$
$$
= \min_{\boldsymbol{x}_\bullet \in t\Delta^K} \left\{ \left( -\sum_{r=1}^K \boldsymbol{g}_\bullet[r]\boldsymbol{x}_\bullet[r] + \frac{1}{2}\sum_{r=1}^K \left(\frac{\boldsymbol{x}_\bullet[r]}{\boldsymbol{w}_\bullet[r]}\right)^2 \right) + V_{Q_k}(\boldsymbol{x}_\bullet[k]; \boldsymbol{g}_k, \boldsymbol{w}_k) \right\}.
$$
$$\tag{27}$$

Suppose that for each $k \in [\![K]\!]$, $\lambda_{Q_k}(t; \boldsymbol{g}_k, \boldsymbol{w}_k)$ is piecewise linear and monotonically increasing in $t$. Now we consider the KKT conditions for $\boldsymbol{x}_\bullet$ in Equation (27):

$$
-\boldsymbol{g}_\bullet[k] + \frac{\boldsymbol{x}_\bullet[k]}{\boldsymbol{w}_\bullet[k]^2} + \lambda_{Q_k}(\boldsymbol{x}_\bullet[k]; \boldsymbol{g}_k, \boldsymbol{w}_k) = \lambda_\bullet + \boldsymbol{\mu}[k] \qquad \forall k \in [\![K]\!] \quad \text{(Stationarity)}
$$

$$
\boldsymbol{x}_\bullet \in t \cdot \Delta^K \qquad\qquad \text{(Primal feasibility)}
$$

$$
\lambda_\bullet \in \mathbb{R}, \boldsymbol{\mu} \in \mathbb{R}^d_{\geq 0} \qquad\qquad \text{(Dual feasibility)}
$$

$$
\boldsymbol{\mu}[k]\boldsymbol{x}_\bullet[k] = 0 \qquad \forall k \in [\![K]\!] \quad \text{(Compl. slackness)}
$$

Solving for $\boldsymbol{x}_\bullet[k]$ in the stationarity condition, and using the conditions $\boldsymbol{x}_\bullet[k]\boldsymbol{\mu}[k] = 0$ and $\boldsymbol{\mu}[k] \geq 0$, it follows that for all $k \in [\![K]\!]$

$$
\boldsymbol{x}_\bullet[k] = \boldsymbol{w}_\bullet[k]^2 \left( \lambda_\bullet + \boldsymbol{\mu}[k] + \boldsymbol{g}_\bullet[k] - \lambda_{Q_k}(\boldsymbol{x}_\bullet[k]; \boldsymbol{g}_k, \boldsymbol{w}_k) \right)
$$
$$
= \boldsymbol{w}_\bullet[k]^2 \left[ \lambda_\bullet + \boldsymbol{g}_\bullet[k] - \lambda_{Q_k}(\boldsymbol{x}_\bullet[k]; \boldsymbol{g}_k, \boldsymbol{w}_k) \right]^+. \tag{28}
$$

**Strict monotonicity and piecewise-linearity of $\boldsymbol{x}_\bullet[k]$ as a function of $\lambda_\bullet$.** Given the preliminaries on SMPL functions, it is now immediate to see that $\boldsymbol{x}_\bullet[k]$ is unique as a function of $\lambda_\bullet$. Indeed, note that (28) can be rewritten as

$$
\boldsymbol{x}_\bullet[k] = \left[ (\boldsymbol{w}_\bullet[k]^2)\lambda_\bullet - \boldsymbol{w}_\bullet[k]^2(-\boldsymbol{g}_\bullet[k] + \lambda_{Q_k}(\boldsymbol{x}_\bullet[k]; \boldsymbol{g}_k, \boldsymbol{w}_k)) \right]^+,
$$

which is a fixed-point problem of the form studied in Lemma 13 for $y = (\boldsymbol{w}_\bullet[k]^2)\lambda_\bullet$ and function $f_k$ defined as

$$
f_k(\boldsymbol{x}_\bullet[k]) = \boldsymbol{w}_\bullet[k]^2(-\boldsymbol{g}_\bullet[k] + \lambda_{Q_k}(\boldsymbol{x}_\bullet[k]; \boldsymbol{g}_k, \boldsymbol{w}_k)),
$$

which is clearly SMPL by inductive hypothesis. Hence, the unique solution to the previous fixed-point equation is given by the quasi-SMPL function

$$
g_k : \lambda_\bullet \mapsto \frac{1}{\boldsymbol{w}_\bullet[k]^2} \left[ (\boldsymbol{x}_\bullet[k] + f_k)^{-1}(\lambda_\bullet) \right]^+,
$$

a standard representation of which can be computed in time $O(d + 1)$ by combining the results of Lemmas 8, 10 and 11 given that a standard representation of $\lambda_{Q_k}(t; \boldsymbol{g}_k, \boldsymbol{w}_k)$ of size $d$ is available by inductive hypothesis.

**Strict monotonicity and piecewise-linearity of $\lambda_\bullet$ as a function of $t$.** At this stage, we know that given any value of the dual variable $\lambda_\bullet$, the unique value of the coordinate $\boldsymbol{x}_\bullet[k]$ that solves the KKT system can be computed using the quasi-SMPL function $g_k$. In turn, this means that we can remove the primal variables $\boldsymbol{x}_\bullet$ from the KKT system, leaving us a system in $\lambda_\bullet$ and $t$ only. We now show that the solution $\lambda_\bullet^\star$ of that system is a SMPL function of $t \in [0, +\infty)$.

Indeed, the value of $\lambda_\bullet^\star$ that solves the KKT system has to satisfy the primal feasibility condition

$$t = \sum_{k=1}^{K} \boldsymbol{x}_\bullet[k] = \sum_{k=1}^{K} g_k(\lambda_\bullet).$$

Fix any $t > 0$. The right-hand side of the equation is a sum of quasi-SMPL functions. Hence, from Lemma 9, we have that the right-hand side has a standard representation of size at most $K + \sum_{k=1}^{K} d_k = d$ can be computed in time $O(d \log K)$. Furthermore, from Lemma 12, we have that the $\lambda_\bullet^\star$ that satisfies the equation is unique, and in fact that the mapping $(0, +\infty) \ni t \mapsto \lambda_\bullet^\star$ is SMPL with standard representation of size at most $d$.

**Relating $\lambda_\bullet$ and $\lambda_Q(t; \boldsymbol{g}, \boldsymbol{w})$.** Since $\lambda_\bullet^\star(t)$ is the coefficient on $t$ in the Lagrangian relaxation of (27), it is a subgradient of $V_Q(t; \boldsymbol{g}, \boldsymbol{w})$, and since there is a unique solution, we get that it is the derivative, that is,

$$\lambda_\bullet^\star(t) = \lambda_Q(t; \boldsymbol{g}, \boldsymbol{w})$$

for all $t \in (0, +\infty)$. To conclude the proof by induction, we then need to analyze the case $t = 0$, which has so far been excluded. When $t = 0$, the feasible set $tQ$ is a singleton, and $V_Q(0; \boldsymbol{g}, \boldsymbol{w}) = 0$. Since $V_Q(t; \boldsymbol{g}, \boldsymbol{w})$ is continuous on $[0, +\infty)$, and since $\lim_{t \to 0^+} \lambda_Q(t; \boldsymbol{g}, \boldsymbol{w}) = \lim_{t \to 0^+} \lambda_\bullet^\star(t)$ exists since $\lambda_\bullet^\star(t)$ is piecewise-linear, then by the mean value theorem,

$$\lambda_Q(0; \boldsymbol{g}, \boldsymbol{w}) = \lim_{t \to 0^+} \lambda_\bullet^\star(t),$$

that is, the continuous extension of $\lambda_\bullet^\star$ must be (right) derivative of $V_Q(t; \boldsymbol{g}, \boldsymbol{w})$ in 0. As extending continuously $\lambda_\bullet^\star(t)$ clearly does not alter its being SMPL nor its standard representation, we conclude the proof of the inductive case.

$\square$

Lemma 14 also provides a constructive way of computing the argmin of (25) in polynomial time for any $t \in [0, +\infty)$. To conclude the construction of the proximal oracle, it is then enough to show how to pick the optimal value of $t \in [0, 1]$ that minimizes

$$\min_{\boldsymbol{x} \in [0,1]Q} \left\{ -\boldsymbol{g}^\top \boldsymbol{x} + \frac{1}{2} \sum_{r=1}^{d} \left( \frac{\boldsymbol{x}[r]}{\boldsymbol{w}[r]} \right)^2 \right\} = \min_{t \in [0,1]} V_Q(t; \boldsymbol{g}, \boldsymbol{w}).$$

That is easy starting from the derivative $\lambda_Q(t; \boldsymbol{g}, \boldsymbol{w})$, which is a SMPL function by Lemma 14. Indeed, if $\lambda_Q(0; \boldsymbol{g}, \boldsymbol{w}) \geq 0$, then by monotonicity of the derivative we know that the optimal value of $t$ is $t = 0$. Else, if $\lambda_Q(1; \boldsymbol{g}, \boldsymbol{w}) \leq 0$, again by monotonicity we know that the optimal value of $t$ is $t = 1$. Else, there exists a unique value of $t \in (0, 1)$ at which the derivative of the objective is 0, and such a value can be computed exactly using Lemma 11.

## C   Experimental Results

In this section we provide preliminary experimental results in order to verify our theoretical findings, and in particular the per-player regret bound established in Theorem 4. More specifically, we investigate the behavior of our learning dynamics (LRL-OFTRL) in four standard extensive-form games used in the literature: 2-player and 3-player *Kuhn poker* [Kuhn, 1950]; 2-player *Goofspiel* [Ross, 1971];[7] and the baseline version of (2-player) *Sheriff* [Farina et al., 2019]. From those games, only 2-player Kuhn poker is a zero-sum game. Our findings are summarized in Figure 1.

---

[7]We consider instances of Goofspiel with $r = 3$ cards and *limited information*—the actions of the other player are only observed at the end of the game. Also, we note that the tie-breaking mechanism makes the game general-sum.

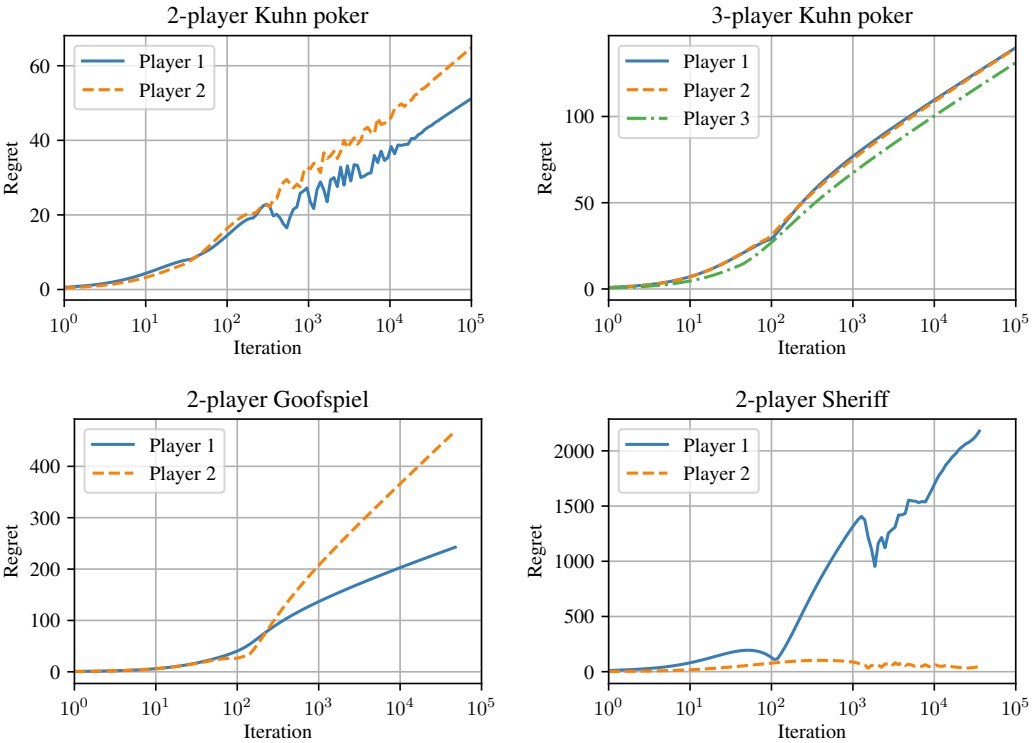

Figure 1: The regret of the players when they follow our learning dynamics, `LRL-OFTRL`; after a very mild tuning process, we selected the same learning rate $\eta \coloneqq 0.5$ for all games. The $x$-axis indexes the iteration, while the $y$-axis the regret. The scale on the $x$-axis is *logarithmic*. We observe that the regret of each player grows as $O(\log T)$, verifying Theorem 4.