# OpenReview forum: "Near-Optimal No-Regret Learning Dynamics for General Convex Games"
_NeurIPS.cc/2022/Conference — NeurIPS 2022 Accept_

### Official Review · Reviewer_S1Jh · 2022-06-21

**Rating:** 7
**Confidence:** 4
**Soundness:** 4 excellent
**Presentation:** 4 excellent
**Contribution:** 3 good

**Summary:**

This paper studies the uncoupled learning for general convex games, which is absolutely of interest to the audience in the community of online learning and game theory. The paper presents novel algorithms equipped with strong regret guarantee in the sense that only poly log T regret is incurred for each player; and also the algorithm can be efficiently implemented given a certain oracle. The crucial novelty in the technical stuff lies in the usage of the lifting to ensure nonengativeity of the regret and also the log barrier function to enhance the stability.

**Questions:**

See above.

**Ethics Review Area:**

["I don’t know"]

**Limitations:**

See above.

**Strengths And Weaknesses:**

The problem studied in this paper is very interesting and challenging. This paper studies the uncoupled learning for general convex games, which is absolutely of interest to the audience in the community of online learning and game theory. The paper presents novel algorithms equipped with strong regret guarantee in the sense that only poly log T regret is incurred for each player; and also the algorithm can be efficiently implemented given a certain oracle. The crucial novelty in the technical stuff lies in the usage of the lifting to ensure nonengativeity of the regret and also the log barrier function to enhance the stability. To the best of my knowledge, there is no similar result of uncoupled learning for convex games even though there is a large body of researches in this thread. The paper writing is of high quality and make the reader feel enjoyable.

The usage of lifting to ensure the nonnegativity is very interesting. Although the lifting idea was used in earlier studies [1,2], using it in uncoupled learning dynamics is very new to the best of my knowledge.

It is a pity that authors fail to cite the two prior works that also use the lifting ideas (in other bandit problems though).
[1] Bias no more: high-probability data-dependent regret bounds for adversarial bandits and MDPs. NeurIPS 2020.

[2] Adaptive Bandit Convex Optimization with Heterogeneous Curvature. COLT 2022.

The authors are suggested to explain more on the lifting operation as stated in Eq. (2). Note that the augmented dimension $\lambda$ also affects the remaining dimensions due to the constraint $y \in \lambda X$, which exhibits salient difference to earlier papers [1,2] that only pave a constant 1 in the augmented dimension.


Using the self-concordant barrier function for full-information game setting is interesting. This will introduce strengthened stability as argued and proven in the paper, while the log barrier function will also make the dimensional dependency worse.


minor comments: Line 191. players need not have any prior --> need not to have

---

> ### Author Response · Authors · 2022-07-29
> **Response to Reviewer S1Jh**
>
> We thank the reviewer for the helpful feedback. Below we address the main question.
>
> --- *“It is a pity that authors fail to cite the two prior works that also use the lifting ideas (in other bandit problems though).”*
>
> Thanks, we will cite the suggested papers that also use a lifting idea. Effectively, the only connection is that we increase the dimension by 1, which is a recurring trick used for very different purposes depending on the application. As the reviewer points out, the key difference is that in the earlier papers the dimension is augmented using a constant value of $1$. On the other hand, our lifting trick is very particular, and ensures that the regret in the lifted space is always nonnegative—which is not true for the lifting used in prior works. We will make sure to include a technical comparison in the revised version to address the reviewer’s comment.

---

### Official Review · Reviewer_SdzV · 2022-07-10

**Rating:** 6
**Confidence:** 4
**Soundness:** 4 excellent
**Presentation:** 3 good
**Contribution:** 2 fair

**Summary:**

The paper considers a convex game (i.e., utility functions that are smooth, and convex given the other actions) between n players, where each player gets the gradient of its utility function at the end of every turn (i.e., full information). An optimistic follow-the-regularized-leader variation is proposed such that if all players use this algorithm with the same tuning, they all achieve a regret of O(logT). If this is not the case, each player can detect that and switch to an algorithm that achieves O(sqrt(T)) against an adversary. The algorithm requires to solve an optimization problem, and the paper proposes two oracles that can provide good approximate solutions to this problem, and analyzes the complexity and the implication for the regret bound.

**Questions:**

1) What do you mean by "efficiently implantable"? especially in light of algorithms like OMWU that seem far more efficient?

2) Is there any concrete motivation that I missed, other than computing the coarse correlated equilibrium? would the regret make sense as a benchmark in some game scenarios?

3) Are there examples for applications where the "full game information" assumption is reasonable?

**Limitations:**

This is a theory pape so there is no societal impact. The technical limitations could be better discussed, as I explain above.

**Strengths And Weaknesses:**

The paper is very well-written and the results are novel and explained well. General convex games are indeed a very broad and interesting class of games. The math is sound and the technical derivations are neat. I'm a bit hesitant regarding the significance or relevance of the results. Some of these issues can be mitigated or at least be presented more accurately, while others are inherent to the type of question the paper is interested in. I do believe that assuming that this question is interesting, the results of the paper are impressive.

The first issue is with the motivation: why would anybody need an algorithm with the (impressive) guarantees proven here. Using the same definition of regret for this game scenario is not obvious to me. It compares the sum of rewards of the players with something completely artificial - the sum of rewards the players would have gotten by playing a fixed action, while all players still follow their sequence of play. However, this sequence of play was generated partly in response to the player's actions. The paper does mention one possible motivation, which is to compute the coarse correlated equilibrium of the game. I think that's a good motivation overall, but if the motivation is computational, it sets some different priorities. Most importantly, I'm not sure if the computational complexity offered by this algorithm makes it a good choice (which I discuss below).

The second issue is with the feedback. Being able to evaluate the gradient in a data-driven scenario is one thing, but being able to do so (with no error) in a game is a different thing. The gradient of player i's utility with respect to its action depends on the actions of other players and this function depends on the structure of the game. I find it a very strong assumption. Even if the algorithm is used to compute a coarse correlated equilibrium (CCE), it's only interesting when the utility functions are unknown - otherwise computing the CCE can be done directly. Providing examples where one easily knows the gradient but not the utility function would help to justify this model, and by that, the motivation of the paper. The examples provided in Section 2.2 are for convex games, and indeed the convexity assumptions are very mild: but the feedback assumption isn't, and I'm not sure how and when it would be satisfied in the given examples. In terms of presentation, the paper claims that the proposed algorithm is an "uncoupled learning algorithm", where uncoupled is defined as "every player is oblivious to the other players’ utilities". First, this definition is soft and vague. Then, it is claimed that the algorithm is even "strongly uncoupled" since players need not have any prior knowledge about the game whatsoever. I don't see how this is the case here if the gradient of the utility function of each player is (indirectly) highly related to the other players' utilities. Is this fair to say the algorithm is uncoupled where all that couples a player to others is given to this player for free, as an arbitrary modeling assumption? Additionally, it seems like the players need to know the Lipschitz parameter L, and the number of players, so I'm not sure if this statement is accurate.

The third issue, which can be easily fixed, is that the paper makes some over statements or just vague technical ones:

1) The statement that the dynamics are "efficiently implementable" is puzzling to me. The number of oracle calls per iteration is T-dependent. Then, each oracle call needs to give an approximate solution with an error of O(1/T), which will require again T-dependent complexity. The proximal oracle requires O(log(log(T)) calls which is nice, but each call will be complicated to implement (providing more detail here is recommended). There, an alternative linear optimization oracle is proposed, but this one needs poly(T) calls per iteration. Now, T needs to be very large to make the regret improvement from O(log^4(T)) significant. In light of this, in what sense is this efficient? numerical simulations could have helped to demonstrate the that algorithm runs in a reasonable time. I find this issue very important if the motivation of the paper is to compute coarse correlated equilibrium. If the overall computation is infeasible, then it's unclear why the  O(log(T)) regret is worst the struggle. I can see why improving the state-of-the-art regret bound for games might require heavy computational machinery like these oracles, but it looks like the paper is claiming that this computational disadvantage is in fact an advantage.

The line about the "best-response oracle" is a bit misleading. Usually, best-response is used in finite discrete games where the "oracle" is just going over the actions. In continuous convex games, best-response amounts to solving a convex problem, so it's not any easier or more common.

Together with the "uncoupling issue" above,  I think that claiming that the algorithm is both uncoupled and efficiently implementable (which is the main presented research question here) is an overstatement (also line 69 and other places).

2) The assumption that the product of the norms, in line 193, is at most 1, is not exactly without the loss of generality. Indeed, one can always rescale the rewards, but as discussed in lines 275-280, this adds a factor to the regret. This factor is recognized in the comparison in Table 1, but not in other places. I don't see why this assumption is needed, instead of just including this factor. This assumption holds for the simplex as the action space, but not, for example, in a multi-dimensional Cournot competition. It's also easy to miss this assumption which is not given near the assumption on the game or the statement of the results, which makes the regret bounds look better than they are in general action spaces.

3) The statement that the algorithm can be adaptive so that if player i is instead facing adversarial utilities the regret is O(sqrt(T)) is a bit vague. The meaning of this adaptivity only becomes clear when reading the proof of the Theorem. But then, it looks like this adaptivity requires each player to know the number of players n, and the Lipschitz parameter L. The step size also requires the player to know these parameters, but at least then one imagine what can be done if they don't know them (or what's the resulting factor in the regret). Since, against an adversary, there are no players at all, this means that the player excepts to play against n-1 other players, and if it finds himself playing against any other number (or players with different tuning), the regret can jump to O(sqrt(T)). I still think that this adaptivity is nice, but I think that the statement in the theorem needs to be more rigorous and include all detail and implicit assumptions.

4) The comparison to Piliouras et al. [2021] is perhaps a bit unfair. Their paper does claim their algorithm is uncoupled, which line 117 here refutes. This seems a very subtle issue given that I'm not sure if the proposed algorithm is uncoupled. I think that discussing how the proposed algorithm is at least "more uncoupled" than Piliouras et al. [2021] is necessary. Then, Table 1 is very confusing with regard to Piliouras et al. [2021]. Where is the T dependence of the regret bound? Also, isn't Piliouras et al. [2021] for finite discrete games, whereas this paper is for continuous games?

I would be happy if I missed some critical details that can mitigate some of these issues. Disconnected from the confusing motivation, the proposed algorithm is creative and neat. Providing some intuition so why and how this lifting and regularizer do the trick can be interesting.

Minor Comments:

The title of Section 2.3 seems inappropriate. This should be the problem formulation.

line 539 - saying that G is defined in (11) can help.

missing period before line 599.

I wasn't able to follow the proof of Proposition 5. What vector is used as the fixed (lambda,y) -star vector here? how come the additional term in the regret bound has the 2 factor, without the 1/(27*eta)?

---

> ### Author Response · Authors · 2022-07-29
> **Response to Reviewer SdzV**
>
> We thank the reviewer for the very detailed feedback. Below we address the concerns.
>
> --- *“[...] It compares the sum of rewards of the players with something completely artificial - the sum of rewards the players would have gotten by playing a fixed action, while all players still follow their sequence of play [...]”*
>
> The reviewer here appears to question whether regret is an appropriate measure of performance in the context of learning in games. But this is at odds with a vast line of prior work that considered special cases of the problem we consider; please see the cited papers. As the reviewer points out, one common motivation revolves around computing CCEs, but more broadly regret is the standard and most common measure of performance in online learning, and in particular when learning in games. We did not attempt to justify the notion of regret in our paper since it is completely standard in the literature.
>
> --- *“The second issue is with the feedback. [...]”*
>
> Using full-information feedback is also a standard assumption in this line of work. We stress that two recent NeurIPS best-paper awards (Syrgkanis et al. (2015), Celli et al. (2020)) also studied no-regret learning in games under full-information feedback; so the broad NeurIPS community clearly finds this line of work important. Of course, we agree with the reviewer that extending this line of work to partial-information models is important, and we believe that our results and framework set the basis for future extensions in general convex games.
>
> --- *“[...] the paper claims that the proposed algorithm is an "uncoupled learning algorithm" [...]”*
>
> We are following the standard notion of uncoupledness from the literature, as introduced by Daskalakis et al. (2011). Each player is adapting only based on gradients **from its own utility function**. No player is assumed to know anything about the utilities of any other player: each player outputs a strategy and receives as input the gradient of their own utility.
>
> --- *“[...] the gradient of the utility function of each player is (indirectly) highly related to the other players' utilities”*
>
> This is not relevant for the definition of uncoupledness used in the literature. Please also see our previous answer.
>
> --- *“it's only interesting when the utility functions are unknown - otherwise computing the CCE can be done directly.”*
>
> Even in settings where the utilities of all players are common knowledge, learning-based approaches are by far the most efficient algorithms for equilibrium computation due to their favorable scalability. So, while the reviewer was probably referring to non-learning methods for computing equilibria—such as the celebrated ellipsoid-against-hope algorithm for computing CCEs in multiplayer games—those algorithms typically don’t scale beyond toy instances. So, we do not agree with the reviewer’s comment that “[learning algorithms are] only interesting when the utility functions are unknown.”
>
> --- *“Additionally, it seems like the players need to know the Lipschitz parameter L, and the number of players [...]”*
>
> This very mild assumption can be bypassed via a standard application of the so-called “doubling trick”—basically a binary search over the learning rate which is now folklore in the literature. Indeed, the number of players and the Lipschitz parameter are only relevant for tuning the (constant) learning rate. Also, this exact assumption is standard in all prior work.
>
> --- *"The statement that the dynamics are "efficiently implementable" is puzzling to me. [...]"*
>
> First of all, the reviewer compares our $O(\log T)$ bounds with the $O(\log^4 T)$ bound of Daskalakis et al., but the latter result only applies to **normal-form games**. On the other hand, the main theme of our paper is about **general convex sets**. It is of course unreasonable to expect a closed-form update rule for a general convex set; instead, every iterate has to be computed using some iterative method, which introduces an iteration dependency on $T$. In light of this, the $O(\log \log T)$ dependence we establish is basically as good as one can hope for in general convex sets. So our dynamics are indeed **efficiently implementable**. Optimizing over a simplex is fundamentally different from optimizing over a general convex set.
>
> To further address the reviewer's concern, we have implemented and tested our algorithm on standard extensive-form games used in the literature. The results verify our theoretical $O(\log T)$ regret bounds; we have included our preliminary experimental results in Appendix C of the revised version, which is visible to the reviewers.
>
> --- *“The assumption that the product of the norms[...]"*
>
> That assumption is without loss of generality, and can be met by simply rescaling utilities, as the reviewer correctly points out. This was just a presentation choice, aimed at reducing the notational burden.
>
> Due to space limitations, we **continue our response below**.

---

> > ### Author Response · Authors · 2022-07-29
> > **Continuation of our Response to Reviewer SdzV**
> >
> > --- *“The statement that the algorithm can be adaptive [...]”*
> >
> > This type of adaptivity against adversarially chosen utilities is standard in this line of work (since at least the first work on the topic, by Daskalakis et al. in 2011), which is why we did not elaborate further. In a nutshell, the adversarial setting is a standard term that means that the player is facing a sequence of utilities produced while the other players are acting so as to maximize the regret of the player. Note that our $O(\log T)$ guarantee for the regret of each player applies when **all** players employ our learning algorithm—that is, each player follows the proposed distributed protocol. On the other hand, in the adversarial regime there are standard lower bounds of $\Omega(\sqrt{T})$ in online learning. So, our main result gives a “best of both worlds” guarantee: We obtain simultaneously near-optimal regret both in the game playing setting, as well as in the adversarial regime.
> >
> > --- “The comparison to Piliouras et al. [2021] is perhaps a bit unfair. [...]“
> >
> > It appears that the reviewer is referring to the second version of the paper of Piliouras et al. that only appeared on arXiv **after the NeurIPS submission deadline**. Our paper correctly claims that their original algorithm was not uncoupled, as Piliouras et al. acknowledge in v2. Our table also accurately reflects the results reported by Piliouras et al. in the original version—which, again, was the only version available to us at the time of the NeurIPS deadline. Our table also points out that the result of Piliouras et al. only applies to normal-form games, as the reviewer perhaps missed. We will update our revised version to accurately reflect the changes made by Piliouras et al.
> >
> > --- *“What do you mean by "efficiently implantable"? [...]”*
> >
> > The reviewer again compares our algorithm for **arbitrary convex sets** to OMWU, which is an algorithm applicable to **simplex domains only**. As we pointed out earlier, it is not reasonable to expect a closed-form update rule for general convex sets. Our algorithm enjoys near-constant-time oracle complexity (specifically, $O(\log \log T)$ proximal oracle calls), and is therefore efficiently implementable.

---

> > > ### Comment · Reviewer_SdzV · 2022-08-06
> > > **Thanks for your response**
> > >
> > > I thank the authors for taking the time to respond to my comments. I'd be happy to focus on the positive aspects of this paper. For this purpose, a more detail-oriented technical response could have been more effective. Perhaps my comments were too scattered. I'll try to reorganize my thoughts in a more unified manner. I'm very open to the possibility I missed some technical aspects, so please prioritize this in your response if possible.
> > >
> > > I'm having difficulties with the "appeal to authority" type of arguments in your response, justifying the regret as a measure and full information as an assumption.  If there's a line of work that uses external regret to measure performance in games, then there should be a good technical reason why. I cannot judge the paper based on how other papers motivated their contribution, which I'm sure they did.  There must be a concrete motivation for thinking about this problem, right? The nice technical results must have some implication, right? clarifying these can help to appreciate the contribution. That's all I'm saying. Right now with the information I have, I can only see computing CCEs as the motivation, which I think is overall a good one. However, the paper is very vague about whether CCEs are the main concern.
> > >
> > > My question about computing the CCE was more basic than that. My understanding is that given the game, computing CCE is just a linear program (LP). Hence, online learning approaches become appealing to compute CCEs when the game is unknown. It would be interesting if the paper could give even a semi-practical example where the proposed approach would be feasible while the LP would not. I do believe such examples should exist, but the issues I see are as follows:
> > >
> > > Issue 1: Full information does typically require some knowledge about the game, which narrows the regime where the proposed approach would be possible while LP would not. Can you give an example where full information would be reasonable even though the game is still unknown?  The examples of section 2.2 are a good start - but they focus on the assumptions of the game itself and not on the full information assumption.
> > >
> > > If there's a good answer to issue (1) above, then I would have to agree that the proposed approach extends the set of games for which the regret approach to compute a CCE is applicable. This already would be nice. The next question would then be how easy this computation is:
> > >
> > > Issue 2: Improving the $T$ dependence of the regret to O(log T) implies that a larger $T$ is required to make the approximation here better than the state of the art, for the special cases where it is applicable. This large $T$ regime conflicts with a $T$ dependent complexity.  The question is then:
> > >
> > > *For games for which existing approaches work (i.e., OMWU for when the action set is a simplex), does the proposed approach have at least comparable complexity?*
> > >
> > > If yes, this is significant. If not, then I don't see how it can be claimed that the approach is efficiently implementable. Of course, I agree that comparing the complexity for general convex action sets is unfair, so I'm only asking about the special cases that have been already analyzed in the literature. Given an answer to Issue (1), I would still agree that the contribution is significant even if the complexity is inferior for these special cases.
> > >
> > >
> > > Some lesser issues:
> > >
> > > 1) I agree that your approach *technically* meets the definition of "uncoupled dynamics", but this is mostly thanks to the full-information assumption. Ad absurdum, if all players have the same reward function, then technically any approach would be uncoupled, but would this be a fair or interesting statement? I'm willing to accept that this issue is subjective. Answering Issue (1) above would make this issue less significant anyway.
> > >
> > > 2) I now understand the issue with Piliouras et al. [2021]. Thanks for clarifying this. Of course, comparing with the newer version is a good idea: it's of course understandable that the papers are concurrent. As I wrote in my original review, the fact that Piliouras et al. [2021]  deal with normal-form games while your work is for continuous games is perhaps the major difference anyway.
> > >
> > > 3) Would your adaptive trick that switches to an adversarial algorithm work without the knowledge of $L$ and the number of players $n$ by using a doubling trick? this is not obvious to me, but I'm willing to believe it given some minimal evidence. If this is not the case, then it would be fair to state that you do need to assume that $L$ and $n$ are known at least for adaptivity. This isn't a major drawback, but it's still some assumption about the game.
> > >
> > > 4) I agree that the assumption about the product of the norms doesn't limit generality, but I think it doesn't reduce any burden since it makes the result harder to read. Lifting this assumption and just giving the general factor in the regret, as you did in Table 1, would be preferable and less confusing.

---

> > > > ### Author Response · Authors · 2022-08-08
> > > > **Response to reviewer's questions**
> > > >
> > > > We thank the reviewer for the detailed response. Below we reply to the main questions.
> > > >
> > > > --- *“If there's a line of work that uses external regret to measure performance in games, then there should be a good technical reason why [...]”*
> > > >
> > > > First, in some special classes of games, such as two-player zero-sum, zero-sum polymatrix, and socially-concave games (Even-dar et al. (2009)), sublinear regret implies convergence to Nash equilibria. More generally, regret minimization implies convergence to coarse correlated equilibria in normal-form games, extensive-form games, and more broadly in continuous games (see Stoltz and Lugosi (2007)); near-optimal rates for general games with convex strategy sets were established in our paper. We view convergence to CCE as a critical motivation for our work, and we will clarify this in the revised version. But more broadly, regret is an intrinsic measure of performance used in online learning. The previous connections make it especially meaningful from a game-theoretic standpoint.
> > > >
> > > > --- *“[...] computing CCE is just a linear program (LP). Hence, online learning approaches become appealing to compute CCEs when the game is unknown.”*
> > > >
> > > > **About “CCE is just a linear program”:** This is not correct, for several reasons:
> > > >
> > > > 1. First of all, even in **multiplayer normal-form games** (let alone general convex games, that is, the focus of this paper) the linear program that computes CCEs has **exponentially many variables** (in the number of players), so it is not “just a linear program”. In extensive-form games (a special case of general convex games), the LP also happens to have **exponentially many constraints** in the game tree size. So, traditional linear programming techniques are not able to compute CCEs in normal-form and extensive-form games in polynomial time in the game description.
> > > >
> > > > 2. In general convex games, such an LP is not known—perhaps one could imagine discretizing into an exponentially large normal-form game, but that would result in a double-exponentially-sized linear program.
> > > >
> > > > 3. In normal-form and extensive-form games, some other methods (specifically the “ellipsoid-against-hope” algorithm by Roughgarden and Papadimitriou (2008) and extensions thereof (Huang and von Stengel (2008)) can compute CCEs by using a particular instantiation of the ellipsoid algorithm (though not as an algorithm to solve a linear program). Our algorithm is significantly more general (applies to any convex game), while at the same time being a decentralized learning algorithm. We remark that even in the two cases where the ellipsoid-against-hope algorithm can be applied, that algorithm is known to scale poorly with the game size.
> > > >
> > > > About **“online learning approaches become appealing to compute CCEs when the game is unknown”**: This is also incorrect.
> > > >
> > > > Learning-based methods are recognized as by far the most scalable and practical approach for equilibrium computation in large games, which is partly the reason why this field of research is so active and focused on regret-based methods.
> > > >
> > > > As a concrete example, consider two-player zero-sum extensive-form games, where computing Nash equilibria can be phrased as an LP of polynomial size. There, LP-based techniques are rarely used beyond small games. Even more, many state-of-the-art regret-based techniques operate in the **full information feedback**. Examples include the counterfactual regret minimization (CFR) algorithm (Zinkevich et al. (2007)), and its modern variants including CFR+ (Tammelin (2014)) and DCFR (Brown and Sandholm (2019)), each of which uses full information feedback and achieved state of the art performance at that time.
> > > >
> > > > So, to summarize, online learning approaches are very appealing and achieve state of the art performance in practice for equilibrium computation, **even when the game is known**, and even in the case where linear programming can be used (we remark again that it is not known how to formulate CCE computation in multiplayer general-sum extensive-form games as a polynomially-sized linear program in the game tree size).
> > > >
> > > > --- *“Issue 1: Full information does typically require some knowledge about the game, which narrows the regime where the proposed approach would be possible while LP would not.”*
> > > >
> > > > The reviewer is mistaken about the applicability of linear programming to CCE computation in general convex games–please see our previous answer. Furthermore, we remark that full information feedback is much weaker than knowing the entire game. For example, it is sufficient that each player knows its own utility function and everyone's strategy at each time. Cournot competition, which is already discussed in our paper, is one example where full information is motivated (see Mertikopoulos and Zhou (2018)).
> > > >
> > > > Extending our result (if possible) to weaker forms of feedback (for example, variants of bandit feedback) is an interesting and challenging direction of research.
> > > >
> > > > **We continue our response below.**

---

> > > > > ### Author Response · Authors · 2022-08-08
> > > > > **Continuation of our response**
> > > > >
> > > > > --- *“For games for which existing approaches work (i.e., OMWU for when the action set is a simplex), does the proposed approach have at least comparable complexity?”*
> > > > >
> > > > > Our algorithm requires $O(\log \log T)$ proximal oracle calls, and in the special case of a $d$-dimensional simplex the proximal oracle can be computed in nearly-linear time $O(d \log d)$. This is only marginally worse than the complexity of OMWU, which is linear in $d$.
> > > > >
> > > > > Also, we disagree with the reviewer’s implication that we are only allowed to claim efficient implementation if our complexity is comparable to OMWU in the special case of a simplex.
> > > > >
> > > > > --- “I agree that your approach technically meets the definition of "uncoupled dynamics", but this is mostly thanks to the full-information assumption”
> > > > >
> > > > > The definition of uncoupled learning dynamics was introduced in the full-information setting in the first place (Hart and Mas-Colell (2000), Daskalakis et al. (2011)), so we don’t understand the use of the word “technically” here. In the special case where players have the same reward functions the notion of uncoupledness is indeed less meaningful, but this is not relevant for the problem in general.
> > > > >
> > > > > -- *“I agree that the assumption about the product of the norms doesn't limit generality, [...]”*
> > > > >
> > > > > We agree with the reviewer on this point. We will make sure to revise that choice.
> > > > >
> > > > > References
> > > > >
> > > > > Even-dar et al. (2009):  On the convergence of regret minimization dynamics in concave games.
> > > > >
> > > > > Stoltz and Lugosi (2007): Learning correlated equilibria in games with compact sets of strategies.
> > > > >
> > > > > Papadimitriou and Roughgarden (2008): Computing correlated equilibria in multi-player games.
> > > > >
> > > > > Huang and von Stengel (2008): Computing an Extensive-Form Correlated Equilibrium in Polynomial Time.
> > > > >
> > > > > Zinkevich et al. (2007): Regret Minimization in Games with Incomplete Information.
> > > > >
> > > > > Tammelin (2014): Solving Large Imperfect Information Games Using CFR+.
> > > > >
> > > > > Brown and Sandholmm (2019): Solving imperfect-information games via discounted regret minimization.
> > > > >
> > > > > Mertikopoulos and Zhou (2018): Learning in Games with Continuous Action Sets and Unknown Payoff Functions.
> > > > >
> > > > > Hart and Mas-Colell (2000): A Simple Adaptive Procedure Leading to Correlated Equilibrium.
> > > > >
> > > > > Daskalakis et al. (2011): Near-Optimal No-Regret Algorithms for Zero-Sum Games.

---

> > > > > > ### Comment · Reviewer_SdzV · 2022-08-08
> > > > > > **Thanks for the informative response!**
> > > > > >
> > > > > > I appreciate your recent detailed technical response which provides important context. I think that revising the paper to provide this context would greatly improve the presentation and  I'm increasing my score assuming such a revision. Some of the details provided in your response are not in the paper yet, and my improved score also assumes they will be added more formally than the rebuttal phase allows. Unfortateunly I cannot check the revised paper, but I trust that the required comparisons with the literature will be fair and avoid overstatements (e.g., reading your response, one might think that the claim that computing a CCE is an LP is not correct).
> > > > > >
> > > > > > The point about online learning being preferable to LP when solving for the CCE is especially important. Discussing this in the paper is therefore crucial. I agree that the LP required to solve the CCE is high-dimensional.  The missing argument is why online learning does a better job than approximate approaches to solve the LP directly (ignoring the game nature of the problem). Comparing the approaches fairly can help to motivate the results of this paper.
> > > > > >
> > > > > > The full-information assumption also becomes more reasonable if the approach is appealing even when the game is known. I agree that if the players know their own utility function and can observe the action profile, they can compute their gradient. My question was if there are examples for distributed scenarios in which this is reasonable to assume. If the algorithm is treated as centralized and used to compute a CCE for a known game, then such an example becomes less crucial, but the writing should make this issue clear.
> > > > > >
> > > > > > The claim that the dynamics are uncoupled is also less important if the algorithm is treated as centralized. By "technically", I meant that if the players know their utility function and can observe the action profile, it is less impressive that the dynamics can be uncoupled. The players already know a lot about how they're coupled with others: they know how other actions affect them, and they can observe these actions. Uncoupled dynamics are more meaningful in a payoff-based case where players can only observe their reward value and nothing else. In the same manner that with a common reward the definition of uncoupled dynamics is not meaningful, then in the full-information scenario, it's just less meaningful. I think that a fair presentation of this claim should mention that.
> > > > > >
> > > > > > The complexity comparison with OMWU for the simplex case is also good news. I couldn't find such a statement in the paper, so improving the discussion in Appendix B, or just elaborating on this special case (e.g., like Proposition 4 does) can be helpful. Again, to be fair, a factor log(log(T))*log(d) compared to OMWU is still somewhat disappointing, but since your algorithm can deal with a general convex action set, this is understandable.

---

> > > > > > > ### Author Response · Authors · 2022-08-09
> > > > > > > **Thank you for the feedback**
> > > > > > >
> > > > > > > Once again, we are grateful to the reviewer for the constructive feedback, and for all the time spent during the reviewing and the discussion periods. We will make sure to incorporate all the suggestions in the revised version.

---

### Official Review · Reviewer_ZFFJ · 2022-07-12

**Rating:** 6
**Confidence:** 3
**Soundness:** 3 good
**Presentation:** 3 good
**Contribution:** 3 good

**Summary:**

This paper studies online learning for general convex games, where the decision set of each player is general convex. The authors propose an algorithm and prove an $O(\log T)$ regret each player, where $T$ is the number of rounds. It improves the best $T^{1/4}$ regret achieved by OMD. At the core of their algorithm is a lifting strategy with a regularized FTRL framework that updates both the utility and the regularization parameter simultaneously.

**Questions:**

- The authors mention that their algorithm will achieve an $\sqrt{T}$ regret under the adversarial setting. Can the authors explain what the adversarial setting is since I do not find the definition in the main text?

- From the algorithm description and the proof, it seems that both the theoretical result and the algorithm can be regarded as those counterparts for a 'single-player' setting, since the regret reg_i cares nothing but the relative optimality of player i, and the algorithm also deals nothing with other players. Can the authors explain more about that?



**Limitations:**

Yes.

**Strengths And Weaknesses:**

Strengths:

- The results are important in theory since it provides the first $\log(T)$ type regret for general convex games.
- The discussion is comprehensive.
- The proof is technically sound.

Weaknesses:
-  There can be more explanation about the problem setting. See questions.

---

> ### Author Response · Authors · 2022-07-29
> **Response to Reviewer ZFFJ**
>
>  We thank the reviewer for the helpful feedback. Below we address the questions raised.
>
> --- *“The authors mention that their algorithm will achieve an $\sqrt{T}$ regret under the adversarial setting. Can the authors explain what the adversarial setting is since I do not find the definition in the main text?”*
>
> The adversarial setting is a standard term that means that the player is facing a sequence of utilities produced while the other players are acting so as to maximize the regret of the player. Note that our $O(\log T)$ guarantee for the regret of each player applies when all players employ our learning algorithm—that is, each player follows the proposed distributed protocol. On the other hand, in the adversarial regime there are standard lower bounds of $\Omega(\sqrt{T})$ in online learning. So, our main result gives a “best of both worlds” guarantee: We obtain simultaneously near-optimal regret both in the game playing setting, as well as in the adversarial regime. We stress that such a consideration has been central in this line of work (e.g., see Syrgkanis et al. (2015)). We will make sure to clarify this point in more detail.
>
> --- *"From the algorithm description and the proof, it seems that both the theoretical result and the algorithm can be regarded as those counterparts for a 'single-player' setting, since the regret reg_i cares nothing but the relative optimality of player i, and the algorithm also deals nothing with other players. Can the authors explain more about that?"*
>
> The reviewer correctly points out that the dynamics we study are fully decentralized and uncoupled, in the sense that each player simply executes its own algorithm based on the observed gradients, without communicating or interacting in any way with the other players. In particular, each player is using a no-regret learning algorithm to adapt to its environment, which in our cases happens to be a multiplayer game. The main challenge in the multi-agent setting is that the observed gradients change depending on the strategies of the other players, so the learning of all the agents makes the environment non-stationary and precludes single-agent approaches.
>
> The existence of fully decentralized and uncoupled learning algorithms that guarantee strong properties in multi-agent settings is a celebrated result at the core of this area of research. Please see our related work section for an overview as to how our work relates and strengthens the rich existing literature on the subject.
>
> We hope that the answers above fully address the reviewer’s concern regarding the lack of explanation about our setting—if not, please let us know, and we can explain further.

---

> > ### Author Response · Authors · 2022-08-08
> > **Thanks for your feedback. Have we addressed the concerns?**
> >
> > We thank again the reviewer for the helpful feedback. Given that the discussion period is soon coming at an end, please let us know if we have adequately addressed the concern regarding lack of explanation about the problem setting, and if the reviewer has any further questions.

---

### Official Review · Reviewer_v5uW · 2022-07-12

**Rating:** 7
**Confidence:** 1
**Soundness:** 3 good
**Presentation:** 4 excellent
**Contribution:** 4 excellent

**Summary:**

This paper proposes a new novel algorithm named LRL-OFTRL which can achieve near-optimal for general convex games. In particular, when all players employ LRL-OFTRL, the regret of each player grows in logarithm speed. Their results significantly extend the result derived by Daskalakis et al. [2021] and can be applied to many subareas such as normal-form games, extensive-form games, splittable routing games, and Cournot competition.

**Questions:**

Since I am not familiar with the area of convex games, I have no technical criticism for this paper.



**Limitations:**

The authors have addressed their work's limitations and potential negative social impact.

**Strengths And Weaknesses:**


Originality: This paper proposes a new novel algorithm named LRL-OFTRL which can achieve near-optimal for general convex games.

Quality: The theory is technically solid and is supported by proof. This is complete work.

Clarity: this paper is well written and organized. The main theorems are supported by proof, but I

Significance: This paper proposes a new algorithm with a novel technique that significantly extends and improves the prior works. I think it is novel.

There are no experiments in this paper. It would make the result more significant if the authors could add some experiments (simulated and/or real data), which (1). shows that the proposed algorithm is implementable, and (2). if possible, show the advantage compared with Kernelized OMWU [Farina et al., 2022].

---

> ### Author Response · Authors · 2022-07-29
> **Response to Reviewer v5uW**
>
> We thank the reviewer for the helpful feedback. Below we address the reviewer’s comments.
>
> --- *“There are no experiments in this paper. It would make the result more significant if the authors could add some experiments (simulated and/or real data), which (1). shows that the proposed algorithm is implementable”*
>
> To address the reviewer’s concern, we have implemented and tested our algorithm on several standard extensive-form games used in the literature. The results verify our theoretical $O(\log T)$ regret bounds. We have included our preliminary experimental results in Appendix C of the revised version, which is visible to the reviewers.
>
> --- *“...and (2). if possible, show the advantage compared with Kernelized OMWU [Farina et al., 2022].”*
>
> Regarding comparison with Kernelized OMWU, one important issue is that the (best known) theoretically sound value for the learning rate required by KOMWU theory is so small (with a leading constant of roughly $10^{-8}$, to be further divided by a $\log^4(T)$ factor) that the regret of KOMWU basically increases linearly in the experiments; this is less of an issue with our algorithm since our analysis gives a much sharper bound for the learning rate. So, under theoretically sound parameterizations we suspect that our method will perform better.
>
> On the other hand, if we allow arbitrary—potentially not theoretically sound—parameterizations, then KOMWU seems to perform better. Nonetheless, KOMWU applies to extensive-form games, while our algorithm is more general in that it applies to arbitrary convex sets. So their guarantees are not quite comparable.

---

> > ### Author Response · Authors · 2022-08-09
> > **Thank you for the feedback. Have we addressed the concerns?**
> >
> > We thank again the reviewer for the helpful feedback. Given that the discussion period is soon coming at an end, please let us know if the experiments we added in the revised version address the reviewer's concerns, and if the reviewer has any further questions or suggestions.

---

### Meta-Review · Area_Chair_fvQA · 2022-08-25

**Recommendation:** Accept
**Confidence:** Certain

**Metareview:**

Reviewers are all positive and appreciate the theoretical contributions. Good work! Please make sure you address all the reviewers' comments and incorporate them (and any new experimental results, if applicable) in your camera-ready.

**Award:**

No

---

### Decision · Program_Chairs · 2022-09-14

Accept